# Learning to Generate Columns with Application to Vertex Coloring

**Yuan Sun**[*]
La Trobe University

**Andreas T. Ernst**
Monash University

**Xiaodong Li**
RMIT University

**Jake Weiner**
RMIT University

## Abstract

We present a new column generation approach based on Machine Learning (ML) for solving combinatorial optimization problems. The aim of our method is to generate high-quality columns that belong to an optimal integer solution, in contrast to the traditional approach that aims at solving linear programming relaxations. To achieve this aim, we design novel features to characterize a column, and develop an effective ML model to predict whether a column belongs to an optimal integer solution. We then use the ML model as a filter to select high-quality columns generated from a sampling method and use the selected columns to construct an integer solution. Our method is computationally fast compared to the traditional methods that generate columns by repeatedly solving a pricing problem. We demonstrate the efficacy of our method on the vertex coloring problem, by empirically showing that the columns selected by our ML model are significantly better, in terms of the integer solution that can be constructed from them, than those selected randomly or based only on their reduced cost. Further, we show that the columns generated by our method can be used as a warm start to boost the performance of a column generation-based heuristic.

## 1 Introduction

Machine Learning (ML) has been increasingly used to tackle combinatorial optimization problems (Bengio et al., 2020), such as learning to branch in Mixed Integer Program (MIP) solvers (Khalil et al., 2016; Balcan et al., 2018; Gupta et al., 2020), learning heuristic search algorithms (Dai et al., 2017; Li et al., 2018), and learning to prune the search space of optimization problems (Lauri & Dutta, 2019; Sun et al., 2021b; Hao et al., 2020). Given a large amount of historical data to learn from, ML techniques can often outperform random approaches and hand-crafted methods typically used in the existing exact and heuristic algorithms.

Predicting optimal solutions for combinatorial optimization problems via ML has attracted much attention recently. A series of studies (Li et al., 2018; Lauri & Dutta, 2019; Sun et al., 2021b; Grassia et al., 2019; Fischetti & Fraccaro, 2019; Lauri et al., 2020; Sun et al., 2021a; 2022; Ding et al., 2020; Abbasi et al., 2020; Zhang et al., 2020) have demonstrated that predicting optimal solution values for individual variables can achieve a reasonable accuracy. The predicted solution can be used in various ways (e.g., to prune the search space of a problem (Lauri & Dutta, 2019; Sun et al., 2021b; Hao et al., 2020) or warm-start a search method (Li et al., 2018; Zhang et al., 2020; Sun et al., 2022)) to facilitate the solving of combinatorial optimization problems.

However, for a symmetric optimization problem, predicting optimal values for individual decision variables does not provide much benefit for solving the problem. For example, in the vertex coloring problem (VCP) (See Section 2 for a formal problem definition), a random permutation of the colors in an optimal solution results in an alternative optimal solution, and thus predicting the optimal colors for individual vertices is not very useful. On the other hand, predicting a complete optimal solution for a problem directly is too difficult. This is partially due to the NP-hardness of a problem and the difficulty in designing a generic representation for solutions of different sizes.

In this paper, we take an intermediate step by developing an effective ML model to predict columns (or fragments (Alyasiry et al., 2019)) that belong to an optimal solution of a combinatorial optimization problem. To illustrate our method, we use the VCP as an example, in which a column is a

---

[*]Corresponding author: yuan.sun@latrobe.edu.au

Maximal Independent Set (MIS) (Tarjan & Trojanowski, 1977), whose vertices can share the same color in a feasible solution. The aim of our ML model is to predict which MISs belong to an optimal solution for a given problem instance.

To train our ML model, we construct a training set using solved problem instances with known optimal solutions, where each training instance corresponds to a column in a training graph. Three categories of features are designed to represent a column, including 1) problem-specific features computed from the graph data, 2) statistical measures computed from sample solutions, and 3) linear program (LP) features (e.g., reduced cost) computed from the LP relaxation of the MIP model. A training instance is labeled as positive if the corresponding column belongs to an optimal solution; otherwise it is labeled as negative. This is then a standard binary classification problem and any existing classification algorithm can be used for this task.

We use the trained ML model to evaluate the quality of columns, and combine it with a sampling method to generate high-quality columns for unseen problem instances. Specifically, our method starts by randomly generating a subset of columns. It then computes the features for each column in the current subset and uses the trained ML model to evaluate the quality of the columns. The low-quality columns predicted by ML are replaced by the new ones generated by a sampling method. This process is repeated for multiple iterations with low-quality columns filtered out and high-quality columns remaining. A subproblem formed by the selected columns is then solved to generate an integer solution. We call this method Machine Learning-based Column Generation (MLCG).

Our MLCG method is a significant departure from the traditional CG methods (Lübbecke & Desrosiers, 2005; Mehrotra & Trick, 1996). Firstly, the traditional methods typically generate columns to solve the LP relaxation of the MIP, while our MLCG method aims at generating columns that are included in a high-quality *integer* solution. Secondly, the traditional methods select columns only based on their reduced cost, while our method learns a more robust criterion via ML based on a set of features for selecting columns. Thirdly, the traditional methods typically generate columns by repeatedly solving a pricing problem, while our method samples columns on the fly and uses the trained ML model to filter out low-quality columns.

To demonstrate the effectiveness of our MLCG method, we evaluate it on the VCP, though the same idea is generally applicable to other combinatorial optimization problems. We empirically show that our ML model can achieve a high accuracy in predicting which columns are part of an optimal solution on the problem instances considered. The columns selected by our ML model are significantly better, in terms of the integer solution that can be constructed from them, than those selected randomly or based purely on their reduced cost. Furthermore, we use the subset of columns generated by our method to warm-start a CG-based heuristic, the Restricted Master Heuristic (RMH) (Taillard, 1999; Bianchessi et al., 2014), and the results show that our method combined with RMH significantly outperforms RMH alone in terms of both solution quality and run time.

## 2 BACKGROUND AND RELATED WORK

**Vertex Coloring Problem Formulation.** Given an undirected graph $G(V, E)$, where $V$ is the set of vertices and $E$ is the set of edges, the objective of VCP is to assign a color to each vertex, such that the adjacent vertices have different colors and the total number of colors used is minimized. Since adjacent vertices cannot share the same color by the problem definition, the vertices that are of the same color in any feasible solution must form an Independent Set. Therefore, the VCP is equivalent to a set partitioning problem which aims to select the minimum number of Independent Sets from a graph such that each vertex is covered *exactly* once. This is also equivalent to a set covering problem, the objective of which is to minimize the number of Maximal Independent Sets (MISs) selected such that each vertex in the graph is covered *at least* once (Mehrotra & Trick, 1996).

Let $S$ denote a MIS, $\mathbb{S}$ denote the set of all MISs of a graph $G$, and $\mathbb{S}_v$ denote the set of MISs that contain vertex $v \in V$. We use a binary variable $x_S$ to denote whether a MIS $S$ is selected. The set covering formulation of VCP is defined in (1)-(3) (Mehrotra & Trick, 1996). A variable $x_S$ corresponds to a MIS in the graph and also a column of the constraint matrix of the MIP. As the number of MISs in a graph is potentially exponential in $|V|$, the number of columns of the MIP can be very large. It can be shown that the LP relaxation of (1)-(3)

$$\min_{\boldsymbol{x}} \sum_{S \in \mathbb{S}} x_S, \qquad (1)$$

$$s.t. \sum_{S \in \mathbb{S}_v} x_S \geq 1, \quad v \in V; \quad (2)$$

$$x_S \in \{0, 1\}, \quad S \in \mathbb{S}. \quad (3)$$

provides a bound at least as good as that of the compact MIP formulation (Mehrotra & Trick, 1996). Although the set covering formulation has a tight LP relaxation, solving the large LP relaxation is difficult. The traditional CG and branch-and-price (B&P) algorithms (Mehrotra & Trick, 1996; Barnhart et al., 1998; Gualandi & Malucelli, 2012) resolve this by generating a subset of columns on the fly, instead of enumerating all the columns upfront, which will be described in the following.

**Column Generation.** Let $\tilde{\mathbb{S}} \subset \mathbb{S}$ denote an arbitrary subset of MISs based on which at least one feasible LP solution can be generated, and $\tilde{\mathbb{S}}_v$ denote the set of MISs in $\tilde{\mathbb{S}}$ that contain vertex $v \in V$. To solve the LP relaxation of (1)–(3), CG starts by solving a *restricted master problem* (4)-(6) with $\tilde{\mathbb{S}}$. The *dual* solution for constraint (5) can be obtained after solving the restricted

$$\min_{\boldsymbol{x}} \sum_{S \in \tilde{\mathbb{S}}} x_S, \qquad (4)$$

$$s.t. \sum_{S \in \tilde{\mathbb{S}}_v} x_S \geq 1, \qquad v \in V; \quad (5)$$

$$0 \leq x_S \leq 1, \qquad S \in \tilde{\mathbb{S}}. \quad (6)$$

master problem: $\boldsymbol{\pi} = \{\pi_1, \pi_2, \ldots, \pi_{|V|}\}$ ($\boldsymbol{\pi} \geq 0$), which can be used to compute the reduced cost of a variable $x_S$ for any $S \in \mathbb{S}$. Let $\boldsymbol{u} = \{u_1, u_2, \ldots, u_{|V|}\}$ be the binary string representation of a MIS $S$, where $u_i = 1$ if the vertex $v_i \in S$, otherwise $u_i = 0$. The *reduced cost* of $x_S$ (for MIS $S$) is computed as $1 - \sum_{i=1}^{|V|} \pi_i \cdot u_i$. The reduced cost computes the amount by which the objective function coefficient of $x_S$ would have to be reduced before $S$ would be cost-effective to use (i.e., $x_S$ would take a non-zero value in the optimal LP solution). In other words, if the reduced cost of a variable $x_S$ is negative, adding $S$ into the subset $\tilde{\mathbb{S}}$ and solving the restricted master problem again would lead to a better LP solution assuming nondegeneracy. If the reduced cost of $x_S$ is non-negative for every $S \in \mathbb{S}$, adding any other MIS into $\tilde{\mathbb{S}}$ would not improve the LP solution. Hence, the LP has been solved to optimality with the current subset $\tilde{\mathbb{S}}$.

However, as the number of MISs in a graph may be exponentially large, explicitly computing the reduced cost for every decision variable $x_S$ is often impractical. CG resolves this issue by optimizing a *pricing problem* (7)-(9) to compute the minimum reduced cost among all variables. The constraints (8) and (9) ensure that the solution $\boldsymbol{u}$ is a valid MIS in the graph $G$. The objec-

$$\min_{\boldsymbol{u}} 1 - \sum_{i=1}^{|V|} \pi_i \cdot u_i, \qquad (7)$$

$$s.t. \ u_i + u_j \leq 1, \qquad (v_i, v_j) \in E; \quad (8)$$

$$u_i \in \{0, 1\}, \qquad v_i \in V. \quad (9)$$

tive (7) minimizes the reduced cost among all possible MISs in the graph. This is equivalent to solving a maximum weighted independent set problem with the dual value $\pi_i$ being the weight of the vertex $v_i$ (for $i = 1, \cdots, |V|$) (Sakai et al., 2003). If the minimum objective value of the pricing problem is negative, the optimal solution generated $\boldsymbol{u}^*$ (which is a MIS) will be added into $\tilde{\mathbb{S}}$, and the above process is repeated. If the minimum objective value of the pricing problem is zero, the LP relaxation of (1)–(3) has been solved to optimality with the current $\tilde{\mathbb{S}}$, and the optimal objective value of the restricted master problem provides a valid lower bound for (1)–(3).

To generate an integer solution to the MIP (1)–(3), a sub-problem with the subset of MISs $\tilde{\mathbb{S}}$ can be solved by a MIP solver. This approach is known as RMH (Taillard, 1999; Bianchessi et al., 2014). Alternatively, CG can be applied at each node of the branch-and-bound tree to compute an LP bound, resulting in a B&P algorithm (Mehrotra & Trick, 1996; Gualandi & Malucelli, 2012).

A potential issue of the CG approach is that it may require to solve many instances of the pricing problem, which may be NP-hard itself. Further, the columns are generated only based on the reduced cost. Although it guarantees the optimality of the LP relaxation, the integer solution contained in the subset may not be of high quality. This motivates us to use ML to learn a better rule from a set of features (including reduced cost), to predict which columns belong to an optimal *integer* solution.

There have been only a few studies that use ML to improve CG and B&P algorithms. Václavík et al. (2018) developed a regression model to predict an upper bound on the optimal objective value of the pricing problem, which was then used to prune the search space of the pricing problem. Morabit et al. (2020) developed a supervised learning model to select the minimum number of columns that could lead to the best LP solution at each iteration of CG. Shen et al. (2022) developed an ML-based pricing heuristic to boost the solving of pricing problems. These methods are still within the scope of the traditional CG framework, which is significantly different from our method, as explained earlier. The ML model designed in (Furian et al., 2021) was used to configure the variable selection and node selection in a B&P algorithm, but not about to improve CG.

## 3 PREDICTING OPTIMAL COLUMNS

In this section, we develop an effective ML model to predict which columns (i.e., MISs) are part of an optimal solution for the VCP.

### 3.1 FEATURE EXTRACTION

Let $\tilde{\mathbb{S}}$ denote a subset of MISs generated from a graph $G(V, E)$, which contains at least one feasible solution. We extract three categories of features to characterize each MIS in $\tilde{\mathbb{S}}$. Note that if the graph is small, we can easily enumerate all the MISs in it, and $\tilde{\mathbb{S}}$ in this case is the full set of MISs.

**Problem-specific features.** Let $S$ be a MIS in $\tilde{\mathbb{S}}$, and $|S|$ be the size of $S$. The first problem-specific feature designed to characterize $S$ is $f_1(S) = |S|/\max_{S' \in \tilde{\mathbb{S}}} |S'|$. Generally, the larger a MIS is, the more likely it belongs to an optimal solution. This is simply because a larger MIS contains more vertices, and therefore it potentially requires fewer MISs to cover all vertices in a graph. Let $\tilde{\mathbb{S}}_v \subset \tilde{\mathbb{S}}$ denote the set of MISs that contain vertex $v \in V$, and $\alpha_v$ denote the maximum size of MISs that contain vertex $v$: $\alpha_v = \max_{S \in \tilde{\mathbb{S}}_v} |S|$. The ratio $|S|/\alpha_v$ computes the relative 'payoff' of using the set $S$ to cover vertex $v$, compared to using the largest MIS in $\tilde{\mathbb{S}}_v$ that can cover vertex $v$. The next four problem-specific features compute the maximum ($f_2$), minimum ($f_3$), average ($f_4$) and standard deviation ($f_5$) of $|S|/\alpha_v$ across the vertices in $S$. Let $\deg(v)$ be the degree of vertex $v \in V$, i.e., the number of neighbours of $v$, and $\Delta$ be the maximum degree of vertices in $V$: $\Delta = \max_{v \in V} \deg(v)$. The next four features designed to characterize $S$ are the maximum ($f_6$), minimum ($f_7$), average ($f_8$) and standard deviation ($f_9$) of the normalized degree ($\deg(v)/\Delta$) across the vertices in $S$.

**Statistical features.** Let $\boldsymbol{x} \in \{0,1\}^{|\tilde{\mathbb{S}}|}$ be the binary string representation of a sample solution to the VCP, where a binary variable $x_S = 1$ if and only if the corresponding MIS $S$ is in the solution, for each $S \in \tilde{\mathbb{S}}$. We first use the method presented in Appendix A.1 to efficiently generate $n$ sample solutions $\{\boldsymbol{x}^1, \boldsymbol{x}^2, \cdots, \boldsymbol{x}^n\}$, where $x_S^i = 1$ if and only if $S$ is in the $i^{th}$ sample solution. Let $\boldsymbol{y} = \{y^1, y^2, \cdots, y^n\}$ be the objective values of the $n$ solutions. The first statistical feature is an objective-based measure, which accumulates the 'payoff' of using $S$ to construct solutions in terms of the objective values $f_{\text{obm}}(S) = \sum_{i=1}^n x_S^i/y^i$. Because the vertex coloring is a minimization problem, a MIS that frequently appears in high-quality sample solutions is expected to have a larger accumulated score. Let $\boldsymbol{r} = \{r^1, r^2, \cdots, r^n\}$ denote the ranking of the $n$ sample solutions in terms of their objective values. The next statistical feature used is the ranking-based measure originally proposed in (Sun et al., 2021b): $f_{\text{rbm}}(S) = \sum_{i=1}^n x_S^i/r^i$. If $S$ frequently appears in high-quality sample solutions (with a smaller rank), it is more likely to have a larger ranking-based score. The objective-based score and ranking-based score are normalized by their maximum values across $\tilde{\mathbb{S}}$. In addition, we compute the Pearson correlation coefficient and Spearman's rank correlation coefficient between the values of the binary decision variable $x_S$ and the objective values $\boldsymbol{y}$ over the sample solutions. If $\boldsymbol{x}_S$ is highly negatively correlated with $\boldsymbol{y}$, it means the sample solutions containing $S$ generally have a smaller objective value than those not having $S$. We present an efficient method to compute these statsitical features in Appendix A.2.

**Linear programming features.** The LP relaxation of a MIP is typically much more efficient than the MIP itself to solve. Solving the LP relaxation can provide very useful information about the importance of a decision variable. Specifically, we solve the restricted master problem (4)–(6) with the subset of MISs $\tilde{\mathbb{S}}$ using the Gurobi solver. The first LP feature extracted for a MIS $S \in \tilde{\mathbb{S}}$ is its value in the optimal LP solution. The optimal LP solution value of $x_S$ is a good indication of which binary value $x_S$ takes in the optimal integer solution, as shown in Appendix B.2 that the mutual information between the LP and integer solutions is noticeable. In general, if $x_S$ has a fractional value closer to 1 in the optimal LP solution, it is more likely to be 1 in the optimal integer solution to the sub-MIP. The next LP feature extracted for $S \in \tilde{\mathbb{S}}$ is the reduced cost (See Section 2) of the corresponding variable $x_S$. When the restricted master problem is solved to optimality for $\tilde{\mathbb{S}}$, the reduced cost of the decision variable $x_S$ is non-negative for every $S \in \tilde{\mathbb{S}}$. Furthermore, if the value of $x_S$ in the optimal LP solution is greater than 0, the reduced cost of $x_S$ must be 0. In general, the larger the reduced cost of $x_S$, the less cost-effective $S$ is to construct solutions.

The three categories of features are designed carefully, each capturing different characteristics of a MIS. The problem specific features focus on the local characteristics of a MIS such as the number

of vertices that a MIS can cover. Our statistical features are motivated by the observation that many optimization problems have a "backbone" structure (Wu & Hao, 2015). In other words, high-quality solutions potentially share some components with the optimal solution. The goal of our statistical features is to extract the shared components from high-quality solutions. The LP features are based on the LP theory, which is widely used by state-of-the-art MIP solvers. The relevance of each category of features is investigated in Appendix B.2 due to the page limit.

### 3.2 CLASS LABELING

To construct a training set, we need to compute the optimal solutions for a set of problem instances for class labeling. We use small graphs in which all the MISs can be enumerated upfront as our training problem instances. For each training graph $G(V, E)$, we first use the method proposed by (Tsukiyama et al., 1977) to list all MISs ($\mathbb{S}$) in the graph. This method is based on vertex sequencing and backtracking, and has a time complexity of $\mathcal{O}(|V| |E| |\mathbb{S}|)$. An exact solver Gurobi is then used to compute the optimal solutions for the problem instance by solving the MIP formulation (1)–(3). The existing optimal solution prediction approaches in literature typically compute only one optimal solution of an optimization problem to supervise the training of an ML model (Li et al., 2018; Sun et al., 2021b; Ding et al., 2020; Sun et al., 2021a). However, this is insufficient in our case, because there often exist multiple optimal solutions in the VCP. For example, in some of the training graphs used in our experiments, more than 90% of MISs are part of optimal solutions, indicating the existence of multiple optimal solutions. To tackle this, we present in Appendix A.3 a brute-force approach to compute for each MIS in a graph whether it belongs to *any* optimal solution. A MIS is assigned with a class label of 1 if it belongs to *any* optimal solution; otherwise 0.

### 3.3 TRAINING AND TESTING

We use multiple graphs to construct a training set. The training graphs are small, so that all the MISs in the graphs can be enumerated easily (Tsukiyama et al., 1977; Csardi & Nepusz, 2006). Each MIS in a training graph is used as a training instance. The features and class label of a MIS are computed as above. After the training set is constructed, an off-the-shelf classification algorithm can be trained to classify optimal and non-optimal MISs (i.e., classifying whether a MIS belongs to an optimal solution or not). We will test multiple classification algorithms including K-Nearest Neighbor (KNN) (Cover & Hart, 1967), Decision Tree (DT) (Breiman et al.; Loh, 2011), and Support Vector Machine (SVM) (Boser et al., 1992; Cortes & Vapnik, 1995) in our experiments. Given an unseen test problem instance, if all the MISs in the corresponding graph can be enumerated upfront, our ML model can be used as a problem reduction technique to prune the search space of the problem (Lauri & Dutta, 2019; Sun et al., 2021b). However, the number of MISs in a test graph may be exponentially large, and hence it is often impossible to enumerate all MISs upfront, especially for large graphs. To tackle this, we develop a search method in the next section to generate a subset of high-quality MISs (without the need of listing all MISs), guided by our ML model.

## 4 GENERATING COLUMNS

This section describes our MLCG method. Our MLCG method starts with a randomly generated subset of columns (i.e., MISs) $\tilde{\mathbb{S}}$, which contains at least one feasible solution. It then goes through $n_{it}$ number of iterations, and at each iteration it computes the features (designed in Section 3.1) for each MIS $S \in \tilde{\mathbb{S}}$, and uses the trained ML model to evaluate the quality of each MIS. The low-quality MISs in $\tilde{\mathbb{S}}$ predicted by ML are replaced by the newly generated ones. Finally, the MISs remain in the subset $\tilde{\mathbb{S}}$ are returned. In this sense, our MLCG method is significantly different from the traditional CG methods that typically generate columns by repeatedly solving a pricing problem.

**Initializing a subset of MISs.** The initial subset of MISs $\tilde{\mathbb{S}}$ is generated randomly. To increase the diversity of $\tilde{\mathbb{S}}$, we generate an equal number of MISs starting from each vertex $v \in V$. This also ensures that $\tilde{\mathbb{S}}$ contains at least one feasible solution, since all vertices in $V$ can be covered by $\tilde{\mathbb{S}}$. To generate a MIS starting from a vertex $v$, we initialize the set $S$ as $v$ and the candidate vertex set $C$ as the set of vertices excluding $v$ and $v$'s neighbors. The set $S$ is then randomly expanded to a MIS. In each step of expansion, we randomly select a vertex $v_s$ from $C$ and add $v_s$ into $S$. The candidate

vertex set $C$ is then updated by removing $v_s$ and $v_s$'s neighbors (due to the definition of independent set). This process is repeated until the candidate vertex set $C$ is empty.

**Evaluating the quality of MISs.** We use the ML model developed in Section 3 to evaluate the quality of the MISs in the subset $\tilde{\mathbb{S}}$. To do this, we first extract the features designed in Section 3.1 for each MIS in $\tilde{\mathbb{S}}$. Note that we do not need to enumerate all the MISs in a graph, because the features can be computed only based on the subset $\tilde{\mathbb{S}}$. The ML model should be computationally efficient, as the time used in prediction should be counted as part of the total run time in solving the problem. Therefore, we only consider linear SVM and DT here, since KNN and non-linear SVM are slow in our case where the number of training instances is large. Let $\boldsymbol{w}$ be the vector of optimal weights learned by linear SVM. The quality score predicted by linear SVM for a given MIS with a feature vector $\boldsymbol{f}$ can be computed by $p_{\mathrm{SVM}}(\boldsymbol{f}) = \sum_{i=1}^{|\boldsymbol{f}|} w_i f_i$. Generally, a MIS with a larger quality score is of better quality. For a trained DT, we use the proportion of positive training instances in a leaf node as an indication of the quality of MISs belonging to that leaf node. The quality score of a MIS with a feature vector $\boldsymbol{f}$ can be computed as $p_{\mathrm{DT}}(\boldsymbol{f}) = n_1(\boldsymbol{f})/(n_0(\boldsymbol{f}) + n_1(\boldsymbol{f}))$, where $n_1(\boldsymbol{f})$ and $n_0(\boldsymbol{f})$ are the number of positive and negative training instances in the leaf node that $\boldsymbol{f}$ belongs to. The score $p_{\mathrm{DT}}$ is in the range of 0 and 1, with 1 indicating the best quality predicted by DT.

**Updating the MIS subset.** We rank the MISs in $\tilde{\mathbb{S}}$ based on their quality scores predicted by the ML model. The top $\kappa$ percentage of MISs are kept and the remaining low-quality MISs are replaced by newly generated MISs. We investigate the following two different methods to generate new MISs. The first method is a *random* approach. To replace a low-quality MIS $S^*$ whose first vertex is $v^*$, the random approach starts from the same vertex $v^*$ and randomly expands it to a MIS. The random approach has the merit of maintaining the diversity of the MIS subset $\tilde{\mathbb{S}}$, but on the other hand it may require many iterations to generate high-quality MISs. The second method is a *crossover* approach inspired by the Genetic Algorithm (Golberg, 1989), which randomly selects two high-quality MISs to create a new one. More specifically, it initializes the set $S$ as the common vertices of the two high-quality MISs selected, and then randomly expands the set $S$ to a MIS.

**Generating an integer solution.** With the subset of MISs obtained, we can use an existing optimization algorithm such as the Gurobi solver to construct a complete integer solution. To be specific, we can form a subproblem by using the subset of MISs generated by our MLCG method as inputs to the MIP formulation (1)–(3), and solve the subproblem using Gurobi. The size of the subproblem is significantly smaller than the original problem, and thus solving the subproblem is a lot easier. In addition, the integer solution generated from the subproblem is likely to be of high-quality, which will be shown in our experiments. Directly solving the subproblem does not provide any optimality guarantee. Alternatively, we can use the subset of MISs generated by our MLCG method to warm-start RMH. Specifically, we can use the MISs generated by our method as the initial subset of columns for the restricted master problem, and solve the LP relaxation of (1)–(3) using a traditional CG approach (see Section 2). The optimal LP objective value provides a valid lower bound for the objective value of integer solutions. We then form a subproblem with the columns generated by our MLCG method and the traditional CG method, and solve the subproblem using an existing algorithm. By doing this, we can obtain a high-quality integer solution with a valid optimality gap. We will evaluate the efficacy of our MLPR method for seeding RMH in our experiments.

## 5 EXPERIMENTS

This section presents the experimental results to show the efficacy of our MLCG method in generating columns for the VCP. Our algorithms are implemented in C++, and the source code is publicly available at `https://github.com/yuansuny/mlcg.git`. Our experiments are conducted on a high performance computing server with multiple CPUs @2.70GHz, each with 4GB memory.

### 5.1 PREDICTION ACCURACY

We use the first 100 problem instances (g001-g100) from MATILDA (Smith-Miles & Bowly, 2015) to construct a training set. These graphs are evolved using Genetic Algorithm to acquire controllable characteristics such as density, algebraic connectivity, energy, and standard deviation of eigenvalues of adjacency matrix. For each instance ($|V| = 100$), we use the algorithm implemented in the igraph library (Tsukiyama et al., 1977; Csardi & Nepusz, 2006) to enumerate all MISs ($\mathbb{S}$) in the

corresponding graph. The Gurobi solver with the default parameter setting, 4 CPUs and 16GB memory is then used to compute the optimal solution value (i.e., class label) for each $S \in \mathbb{S}$. The constructed training set consists of $94,553$ positive (with a class label 1) and $94,093$ negative (with a class label 0) training instances (i.e., MISs). The training set is reasonably balanced, indicating that there often exist multiple optimal solutions in the VCP. The distribution of the training instances in the 2D feature space created by PCA is presented in Appendix B.1.

We test multiple classifiers for this classification task, including KNN ($k = 3$), DT (depth $= 20$), linear SVM ($C = 1$) and nonlinear SVM with the RBF kernel ($C = 1000$). The implementations of KNN and DT are from the scikit-learn library (Pedregosa et al., 2011); the nonlinear SVM is from the LIBSVM library (Chang & Lin, 2011); and the linear SVM is from the LIBLINEAR library (Fan et al., 2008). The parameters of these classification algorithms are tuned by hand. All the algorithms and datasets used are publicly available. We train each classification algorithm on the training set with fifteen features, and the ten-fold cross-validation accuracy of KNN, DT, linear SVM, and nonlinear SVM are 91%, 90%, 78% and 85%, respectively. The correlation between each feature and the class label, the feature weights learned by the linear SVM, and a trained simple DT are presented and discussed in Appendix B.2.

## 5.2 The Efficacy of Generating MISs

**Test instances.** We use 33 problem instances from MATILDA (Smith-Miles & Bowly, 2015) as our test instances, which cannot be optimally solved by Gurobi (with 4 CPUs) in 10 seconds. The number of vertices in those graphs are all $100$. For each test instance, we use our MLCG method to generate a subset of MISs, and use Gurobi to solve a sub-problem formed by the generated MISs. The quality of the best integer solution found in the sub-problem is an indication of the quality of the MISs generated by our MLCG method.

**Parameter setting.** The size of the MIS subset $|\tilde{\mathbb{S}}|$ is set to $20|V|$, where $|V|$ is the number of vertices in a graph. In general, a larger $|\tilde{\mathbb{S}}|$ leads to a better integer solution, but at an expense of longer runtime. The number of iterations $n_{\text{it}}$ is set to 10 for now, and will be tuned later. In each iteration of our MLCG method, 50% of MISs are replaced: $\kappa = 50$.

**Linear SVM versus Decision Tree.** We test two efficient classification algorithms, linear SVM and DT, for evaluating the quality of MISs. The average optimality gap generated by our MLCG method with linear SVM is 2.49%, which is significantly better than that with DT 7.95% (See Appendix B.3). This indicates that linear SVM is more effective than DT in evaluating the quality of MISs, which is somewhat surprising because the classification accuracy of DT is much higher than that of linear SVM. In the rest of our experiments, we will only use linear SVM to evaluate MISs.

**Random update versus crossover update.** We test two approaches (random and crossover) for generating new MISs to replace low-quality MISs in each iteration of our algorithm. The two approaches perform similarly, with the average optimality gap generated by the random approach 2.49% and that of the crossover approach 2.68% (See Appendix B.4). The reason why the crossover approach is not more effective may be that the diversity of the generated MISs is important. In particular, having more high-quality but similar MISs in the subset is not expected to result in a better integer solution. In the rest of the paper, we simply use the random approach to generate new MISs.

**Number of iterations.** We vary the number of iterations ($n_{\text{it}}$) from 1 to 100, and record the optimality gap generated by our MLCG method. The solution quality generated by our method improves dramatically in the first several iterations, and the improvement slows down later on. The smallest optimality gap is generated at around the $50^{th}$ iteration, and cannot be further reduced afterwards (See Appendix B.5 for the detailed results). Hence, we set $n_{\text{it}} = 50$ hereafter.

**Baselines.** We compare our MLCG method to four baselines for generating columns: *1) Random*, which randomly generates 10 MISs starting from each vertex in a graph, resulting in $10|V|$ in total (the same number as our method $20|V| \times 50\%$); *2) RC*, which replaces linear SVM with a simple criterion of using the reduced cost to evaluate MISs; *3) MLF*, which enumerates all MISs in a graph and uses linear SVM to select the top 10 MISs for each vertex, thus $10|V|$ in total; and *4) Full*, which simply enumerates and uses all MISs in a graph. The MISs generated by each method are used to construct a (sub-)problem, which is then solved by Gurobi with 4 CPUs and a one-hour cutoff time.

Table 1: The experimental results generated by each algorithm for solving the test problem instances. The last row presents the p-value of the $t$-tests between the results generated by our MLCG method and each of the other methods. The ones with statistical significance (p-value $< 0.05$) are in bold.

| Graph | Density | Optimality Gap (%) | | | | | Runtime | | | | |
|---|---|---|---|---|---|---|---|---|---|---|---|
| | | MLCG | MLF | RC | Rand | Full | MLCG | MLF | RC | Rand | Full |
| g134 | 0.34 | 10.00 | 11.20 | 10.00 | 22.40 | 10.00 | 137.04 | 81.27 | 122.99 | 130.11 | 3605.50 |
| g135 | 0.33 | 0.67 | 8.33 | 8.33 | 13.00 | 0.00 | 50.80 | 10.78 | 7.61 | 28.27 | 47.73 |
| g144 | 0.37 | 10.00 | 13.20 | 10.40 | 12.40 | 0.00 | 5.10 | 6.74 | 6.12 | 1.53 | 20.92 |
| g171 | 0.28 | 0.00 | 0.00 | 0.00 | 11.50 | 0.00 | 5.29 | 11.61 | 5.20 | 0.32 | 14.55 |
| g173 | 0.36 | 0.00 | 12.50 | 1.00 | 12.50 | 0.00 | 6.29 | 9.70 | 4.84 | 0.52 | 14.39 |
| g175 | 0.32 | 0.00 | 12.50 | 0.00 | 12.00 | 0.00 | 5.44 | 31.58 | 5.50 | 0.54 | 40.09 |
| g214 | 0.26 | 0.00 | 0.00 | 0.00 | 12.00 | 0.00 | 4.38 | 15.12 | 4.24 | 0.18 | 26.15 |
| g243 | 0.30 | 10.00 | 18.00 | 20.00 | 20.00 | 20.00 | 33.63 | 27.41 | 13.93 | 29.33 | 3611.80 |
| g263 | 0.23 | 0.00 | 20.00 | 13.60 | 17.60 | 0.00 | 5.11 | 5.48 | 4.52 | 0.20 | 20.97 |
| g324 | 0.38 | 0.00 | 0.00 | 0.00 | 5.00 | 0.00 | 6.52 | 17.90 | 4.63 | 0.62 | 25.08 |
| g329 | 0.21 | 0.00 | 4.00 | 1.60 | 6.40 | 0.00 | 4.44 | 8.31 | 4.65 | 0.32 | 16.61 |
| g348 | 0.21 | 0.00 | 0.00 | 0.00 | 10.29 | 0.00 | 4.99 | 9.80 | 4.80 | 0.16 | 17.17 |
| g417 | 0.40 | 2.00 | 12.50 | 1.00 | 24.00 | 0.00 | 5.51 | 12.63 | 5.42 | 1.17 | 12.89 |
| g420 | 0.41 | 0.00 | 0.00 | 0.00 | 0.00 | 0.00 | 4.46 | 7.89 | 6.04 | 0.11 | 107.48 |
| g474 | 0.34 | 11.11 | 22.22 | 14.67 | 22.22 | 0.00 | 6.40 | 21.75 | 12.71 | 1.54 | 489.22 |
| g479 | 0.23 | 0.00 | 0.00 | 0.00 | 6.67 | 0.00 | 4.78 | 18.48 | 4.81 | 0.65 | 34.95 |
| g486 | 0.67 | 0.19 | 0.00 | 3.62 | 4.76 | 0.00 | 21.66 | 20.88 | 20.58 | 3.24 | 11.32 |
| g487 | 0.41 | 0.00 | 0.00 | 0.00 | 8.92 | 0.00 | 20.84 | 13.14 | 13.26 | 31.06 | 864.33 |
| g493 | 0.24 | 0.00 | 0.00 | 1.33 | 10.67 | 0.00 | 4.59 | 4.92 | 5.52 | 0.30 | 12.52 |
| g496 | 0.30 | 7.43 | 0.00 | 1.14 | 3.43 | 0.00 | 6.16 | 7.05 | 4.40 | 0.21 | 46.25 |
| g498 | 0.35 | 0.00 | 0.00 | 3.43 | 12.00 | 0.00 | 4.75 | 14.54 | 4.04 | 0.13 | 45.18 |
| g508 | 0.53 | 0.00 | 0.00 | 0.89 | 0.00 | 0.00 | 3.98 | 9.66 | 5.65 | 0.14 | 10.19 |
| g509 | 0.52 | 8.44 | 11.11 | 6.67 | 10.67 | 0.00 | 4.46 | 20.59 | 3.72 | 0.22 | 31.21 |
| g511 | 0.50 | 0.00 | 0.00 | 0.00 | 0.00 | 0.00 | 4.57 | 10.51 | 4.54 | 0.17 | 15.31 |
| g530 | 0.51 | 0.00 | 0.00 | 0.84 | 8.84 | 0.00 | 6.85 | 14.13 | 6.60 | 0.40 | 15.03 |
| g556 | 0.35 | 0.00 | 0.36 | 0.00 | 14.55 | 0.00 | 94.60 | 72.03 | 84.98 | 144.22 | 2753.70 |
| g557 | 0.56 | 0.00 | 0.00 | 0.00 | 5.65 | 0.00 | 37.01 | 40.85 | 17.67 | 16.26 | 200.79 |
| g597 | 0.46 | 0.00 | 0.00 | 0.00 | 7.14 | 0.00 | 115.78 | 61.61 | 38.31 | 96.75 | 814.56 |
| g598 | 0.48 | 0.00 | 0.57 | 0.00 | 14.00 | 0.00 | 110.11 | 80.31 | 67.73 | 11.59 | 748.36 |
| g617 | 0.29 | 0.00 | 0.00 | 0.00 | 1.50 | 0.00 | 4.15 | 20.98 | 4.15 | 0.16 | 23.40 |
| g876 | 0.43 | 1.71 | 14.29 | 13.71 | 14.29 | 0.00 | 5.33 | 4.48 | 4.77 | 0.34 | 22.78 |
| g894 | 0.43 | 0.00 | 12.57 | 0.57 | 9.14 | 0.00 | 6.26 | 8.24 | 5.28 | 0.41 | 12.10 |
| g913 | 0.44 | 0.00 | 0.00 | 7.43 | 14.29 | 0.00 | 4.38 | 7.45 | 6.25 | 0.18 | 29.17 |
| Mean | | 1.87 | 5.25 | 3.64 | 10.54 | 0.91 | 22.60 | 21.45 | 15.62 | 15.19 | 417.02 |
| P-value | - | - | **2.23e-03** | **1.69e-02** | **5.14e-10** | 1.42e-01 | - | 7.29e-01 | **2.45e-02** | **4.06e-02** | **2.26e-02** |

**Comparison results.** The optimality gap (with respect to the lower bound produced by Full) of each algorithm for each test problem instance and the runtime of the algorithms are presented in Table 1. The results are averaged over 25 independent runs. Comparing our MLCG method with Full (which is an exact approach), we can clearly see that MLCG consistently generates an optimal or near-optimal solution using a much less runtime. On average, our MLCG method reduces 95% of the runtime of Full, and only increases the optimality gap from 0.91% to 1.87%. Notably on g243, the full problem cannot be solved to optimality by Gurobi in a one-hour cutoff time. In contrast, using the columns generated by our MLCG method to construct a sub-problem, Gurobi is able to find a better integer solution in only 33.63 seconds. Compared to Random, our MLCG method consistently generates a better solution for the test problem instances except g496. The average optimality gap generated by our MLCG method is about 5 times better (smaller) than that of Random. Furthermore, our MLCG method significantly outperforms the RC method in terms of the optimality gap generated (with p-value = 0.0169). This clearly demonstrates that our ML model has learned a more useful rule than the reduced cost for evaluating the quality of MISs. Surprisingly, the MLF method does not perform as well as the MLCG method in terms of the optimality gap. The reason for this is that the aim of CG is to select a diverse set of high-quality MISs to cover each vertex at least once. The ML model can evaluate the quality of MISs, but it cannot measure the diversity of MISs selected. In the MLF approach where all MISs are listed, multiple high-quality but similar MISs (i.e., sharing many common vertices) are likely to be selected. Although those selected MISs are individually of high-quality, when combined it loses diversity. Hence, the MLF method does not perform well compared to the MLCG method.

## 5.3 SEEDING RESTRICTED MASTER HEURISTIC

We use the columns generated by our MLCG method to warm-start RMH, namely MLRMH, compared to RMH with the typical random initialization. The number of initial MISs is set to $10|V|$. The restricted master problem and the pricing problem are both solved by Gurobi. Apart from the 33 MATILDA problem instances used above, we include 30 other MATILDA problem instances, in

Table 2: The comparison between the MLRMH and RMH algorithms for solving the three sets of test problem instances. The results are averaged across the instances in each problem set.

| Graphs | Optimality Gap (%) | | CG Iterations | | LP Runtime | | Total Runtime | |
|---|---|---|---|---|---|---|---|---|
| | MLRMH | RMH | MLRMH | RMH | MLRMH | RMH | MLRMH | RMH |
| MATILDA (33) | 3.89 | 8.21 | 7.08 | 69.63 | 9.25 | 49.61 | 28.08 | 74.35 |
| MATILDA (30) | 8.96 | 12.53 | 59.38 | 156.28 | 37.39 | 74.77 | 58.30 | 84.99 |
| DIMACS (12) | 24.56 | 27.03 | 84.21 | 168.54 | 198.34 | 656.74 | 12439.00 | 12668.00 |

which the number of MISs is so large that they cannot be enumerated with 16GB memory. In addition, we evaluate the generalization of our method on a set of larger DIMACS problem instances, in which the number of vertices vary from 125 to 1000.

The experimental results are summarized in Table 2, and the detailed experimental settings and results are shown in Appendix B.6. Note that the optimal LP objective obtained is a valid lower bound (LB) for the objective value of integer solutions, and is used to compute the optimality gap. The LB generated by MLRMH (or RMH) is very tight for most of the problem instances tested. Our MLRMH algorithm is able to prove optimality for many of the problem instances and provide an optimality gap for the remainder, making it a quasi-exact method. The average optimality gap generated by MLRMH is significantly smaller than that of RMH. Interestingly, by using the columns generated by our MLCG method to warm-start RMH, the number of CG iterations is substantially reduced. This also leads to a significant reduction in the runtime of solving the LP. Note that the time used by our MLCG method to generate columns is counted as part of the LP solving time.

Our method is not confined to solving problem instances generated from a similar distribution. In fact, the MATILDA graphs we used are evolved using Genetic Algorithm to acquire various characteristics such as density, algebraic connectivity, energy, and standard deviation of eigenvalues of adjacency matrix etc (Smith-Miles & Bowly, 2015), and the DIMACS graphs are a collection of hard problem instances from multiple resources with very different problem characteristics and sizes. Our experimental results showed that our method trained on a subset of MATILDA graphs performs well across the MATILDA and DIMACS graphs.

# 6 CONCLUSION

In this paper, we have developed an effective CG method based on ML for the VCP. We defined a column as a Maximal Independent Set whose vertices can share the same color in a feasible solution, and developed an ML model to predict which column belongs to an optimal solution. Novel features were designed to characterize a column, including those computed from the graph data, sample solutions and the LP relaxation. We then used the ML model to evaluate the quality of columns, and incorporated it into a search method to generate a subset of high-quality columns without the need of enumerating all columns in a problem instance. We empirically showed that our ML model achieved a high accuracy in predicting which columns are optimal on the datasets considered. Furthermore, we showed that the columns generated by our method were significantly better than those generated randomly or via the reduced cost. Finally, we showed that the columns generated by our method can be used to boost the performance of a CG-based heuristic as a warm start.

Our proposed MLCG method is generic. Although we have only demonstrated the efficacy of this approach on the VCP, the same idea can be applied to other combinatorial optimization problems. Taking the vehicle routing problem as an example, in which a column is a resource constrained shortest path, an ML model can be built to predict which path is likely to be part of the optimal solution. This will require the design of problem specific features to characterize a path and a specialized sampling method to efficiently generate paths. Our paper has focussed on using CG as part of a heuristic approach, where only a restricted master problem is solved to obtain a final integer solution, as is commonly found in many papers applying CG to large scale problems. However, the same approach could easily be applied in the context of an exact B&P method, where CG would be done predominantly using the ML approach with reduced-cost based pricing of columns only completed at the end to prove optimality of the LP at each node. Our results are indicative that such an ML boosted approach may be expected to have a positive impact on the performance of a B&P approach as well. However a rigorous investigation of this is beyond the scope of the current paper.

ACKNOWLEDGEMENT

This work was supported by an ARC Discovery Grant (DP180101170) from Australian Research Council.

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

# A APPENDIX

## A.1 AN EFFICIENT RANDOM SAMPLING METHOD

We describe an efficient random sampling method to generate sample solutions to the VCP in Algorithm A.1. The inputs to the algorithm are the graph $G(V, E)$ and the set of MISs $\tilde{\mathbb{S}}_v$ containing vertex $v$, for each $v \in V$. The output is a set of MISs selected which can cover all the vertices in $V$ at least once (i.e., a feasible solution denoted as $\mathbb{S}_x$). The random sampling method starts by initializing the solution $\mathbb{S}_x$ as empty and marking each vertex $v \in V$ as not covered. It then iterates through each vertex $v \in V$, and if $v$ has not been covered, it randomly selects a MIS $S$ from $\tilde{\mathbb{S}}_v$ and adds $S$ to the solution $\mathbb{S}_x$. The vertices in $S$ are then marked as covered. A feasible solution $\mathbb{S}_x$ is obtained when all vertices in $V$ have been covered.

---

**Algorithm A.1** Random Sampling

---

**Input:** graph $G(V, E)$, MISs sets $\tilde{\mathbb{S}}_v$ for each $v \in V$.
**Output:** a sample solution $\mathbb{S}_x$.
Initialize $\mathbb{S}_x \leftarrow \{\}$.
Initialize $covered[v] \leftarrow 0$, for each $v \in V$.
**for each** $v \in V$ **do**
    **if** $covered[v] = 0$ **then**
        randomly select a MIS $S$ from $\tilde{\mathbb{S}}_v$.
        add $S$ into $\mathbb{S}_x$.
        **for each** $\tilde{v} \in S$ **do**
            $covered[\tilde{v}] \leftarrow 1$.

---

**Lemma A.1.** *Given a graph $G(V, E)$, the time complexity of generating a sample solution to the VCP using Algorithm A.1 is $\mathcal{O}(|V|)$ in the best case, and $\mathcal{O}(\alpha|V|)$ in the worst case, where $\alpha = \max_{S \in \tilde{\mathbb{S}}} |S|$ is the maximum size of the MISs in $\tilde{\mathbb{S}}$.*

*Proof.* In the best case, the MISs selected do not share any common vertex. In other words, each vertex in the graph is covered exactly once. In this case, the number of basic operations (e.g., comparison and assignment) performed in Algorithm A.1 is about $3|V|$, which is in $\mathcal{O}(|V|)$. In contrast, the MISs selected may share many common vertices and each vertex in the graph may be covered multiple times. In the worst case, each MIS selected in Algorithm A.1 can only cover one new vertex that has not been covered (with other vertices in the MIS already being covered). In this case, the number of assignments performed in the inner loop of Algorithm A.1 is $\sum_{i=1}^{|V|} |S_i|$, where $|S_i|$ is the size of the $i^{th}$ selected MIS. As $\alpha$ denotes the maximum size of the MISs in $\tilde{\mathbb{S}}$, we have $|S_i| \leq \alpha$ and $\sum_{i=1}^{|V|} |S_i| \leq \alpha|V|$. Hence the worst case time complexity of Algorithm A.1 is $\mathcal{O}(\alpha|V|)$. $\qquad\square$

We run Algorithm A.1 $n$ times to generate $n$ sample solutions $\{\mathbb{S}_x^1, \mathbb{S}_x^2, \cdots, \mathbb{S}_x^n\}$. In each run, we randomly permute the vertex set $V$ to increase the diversity of the generated solutions. The sample size $n$ should be large enough so that each MIS in $\tilde{\mathbb{S}}$ is expected to be sampled at least once. In our experiments, we set $n = |\tilde{\mathbb{S}}|$.

## A.2 AN EFFICIENT METHOD FOR COMPUTING STATISTICAL FEATURES

We first describe the statistical features in more detail. Let $\boldsymbol{x} \in \{0, 1\}^{|\tilde{\mathbb{S}}|}$ be the binary string representation of a sample solution, where a binary variable $x_S = 1$ if and only if the corresponding MIS $S$ is in the solution, for each $S \in \tilde{\mathbb{S}}$. The $n$ sample solutions can be denoted as $\{\boldsymbol{x}^1, \boldsymbol{x}^2, \cdots, \boldsymbol{x}^n\}$, where $x_S^i = 1$ if and only if $S$ is in the $i^{th}$ sample solution. Let $\boldsymbol{y} = \{y^1, y^2, \cdots, y^n\}$ be the objective values of the $n$ solutions. The first statistical feature is an objective-based measure, which accumulates the 'payoff' of using $S$ to construct solutions in terms of the objective values:

$$f_{\text{obm}}(S) = \sum_{i=1}^{n} x_S^i / y^i. \tag{10}$$

Because the vertex coloring is a minimization problem, a MIS that frequently appears in high-quality sample solutions (with smaller objective values) is expected to have a larger accumulated score. As the scale of the objective-based score is sensitive to the scale of objective values and sample size, we normalize the objective-based scores by the maximum objective-based score in $\tilde{\mathbb{S}}$: $f_{10}(S) = f_{\text{obm}}(S)/\max_{S' \in \tilde{\mathbb{S}}} f_{\text{obm}}(S')$.

Let $\boldsymbol{r} = \{r^1, r^2, \cdots, r^n\}$ denote the ranking of the $n$ sample solutions in terms of their objective values. The next statistical feature used is the ranking-based measure originally proposed by (Sun et al., 2021b):

$$f_{\text{rbm}}(S) = \sum_{i=1}^{n} x_S^i / r^i. \tag{11}$$

If $S$ frequently appears in high-quality sample solutions (with a smaller rank), it is more likely to have a larger ranking-based score. The ranking-based score is normalized by the maximum ranking-based score in $\tilde{\mathbb{S}}$: $f_{11}(S) = f_{\text{rbm}}(S)/\max_{S' \in \tilde{\mathbb{S}}} f_{\text{rbm}}(S')$.

The Pearson correlation coefficient between the values of the binary decision variable $x_S$ and the objective values $\boldsymbol{y}$ over the sample solutions can also be used to quantify the expected benefits of using $S$ to construct solutions:

$$f_{12}(S) = \frac{\sum_{i=1}^{n}(x_S^i - \bar{x}_S)(y^i - \bar{y})}{\sqrt{\sum_{i=1}^{n}(x_S^i - \bar{x}_S)^2}\sqrt{\sum_{i=1}^{n}(y^i - \bar{y})^2}}, \tag{12}$$

where $\bar{x}_S = \sum_{i=1}^{n} x_S^i/n$ and $\bar{y} = \sum_{i=1}^{n} y^i/n$. If $\boldsymbol{x}_S$ is highly negatively correlated with $\boldsymbol{y}$, it means the sample solutions containing $S$ generally have a smaller objective value than those not having $S$. Thus, using $S$ to construct solutions is expected to result in a smaller objective value.

The last statistical measure computes the Pearson correlation coefficient between the values of the binary decision variable $x_S$ and the ranking of the sample solutions ($\boldsymbol{r}$):

$$f_{13}(S) = \frac{\sum_{i=1}^{n}(x_S^i - \bar{x}_S)(r^i - \bar{r})}{\sqrt{\sum_{i=1}^{n}(x_S^i - \bar{x}_S)^2}\sqrt{\sum_{i=1}^{n}(r^i - \bar{r})^2}}, \tag{13}$$

where $\bar{x}_S = \sum_{i=1}^{n} x_S^i/n$ and $\bar{r} = \sum_{i=1}^{n} r^i/n$. This ranking-based correlation score is equivalent to Spearman's rank correlation coefficient between $\boldsymbol{x}_S$ and $\boldsymbol{y}$, as the value of $x_S^i$ can be interpreted as its rank (considering tied ranks). A MIS with a high negative ranking-based correlation score is more likely to appear in high-quality sample solutions.

Computing the statistical features based on the binary string representation of sample solutions ($\boldsymbol{x}$) can be achieved in $\mathcal{O}(n|\tilde{\mathbb{S}}|)$. This is computationally slow when the number of samples ($n$) and the number MISs ($|\tilde{\mathbb{S}}|$) are large. Based on the fact that $\boldsymbol{x}$ is a vector of binary variables, we introduce an efficient method to compute the statistical features in the following.

**Lemma A.2** ((Sun et al., 2021b)). *For a binary variable $x_S^i$, the following equality holds:*

$$\sigma_S^x = \sum_{i=1}^{n}(x_S^i - \bar{x}_S)^2 = n\bar{x}_S(1 - \bar{x}_S), \tag{14}$$

*where $\bar{x}_S = \sum_{i=1}^{n} x_S^i/n$.*

*Proof.* Here, we provide a different proof to the Lemma than (Sun et al., 2021b).

$$\sigma_S^x = \sum_{i=1}^{n}(x_S^i - \bar{x}_S)^2 = \sum_{i=1}^{n}\left((x_S^i)^2 - \bar{x}_S^2\right) = \sum_{i=1}^{n}(x_S^i)^2 - n\bar{x}_S^2. \tag{15}$$

As $x_S^i$ is a binary variable, $\sum_{i=1}^{n}(x_S^i)^2 = \sum_{i=1}^{n} x_S^i = n\bar{x}_S$. Hence,

$$\sigma_S^x = n\bar{x}_S - n\bar{x}_S^2 = n\bar{x}_S(1 - \bar{x}_S). \tag{16}$$

$\square$

---

**Algorithm A.2** Computing Statistical Features

---

**Input:** sample solutions $\{\mathbb{S}_x^1, \cdots, \mathbb{S}_x^n\}$, objective values $\{y^1, \cdots, y^n\}$, rankings $\{r^1, \cdots, r^n\}$, MIS subset $\tilde{\mathbb{S}}$.
**Output:** statistical features $f_{\mathrm{obm}}(S)$, $f_{\mathrm{rbm}}(S)$, $f_{12}(S)$ and $f_{13}(S)$, for each $S \in \tilde{\mathbb{S}}$.
Initialize $f_{\mathrm{obm}}(S) \leftarrow 0$, $f_{\mathrm{rbm}}(S) \leftarrow 0$, for each $S \in \tilde{\mathbb{S}}$.
Initialize $\bar{x}_S \leftarrow 0$, $\sigma_S^{xy} \leftarrow 0$, $\sigma_S^{xr} \leftarrow 0$, for each $S \in \tilde{\mathbb{S}}$.
Compute the mean objective value: $\bar{y} \leftarrow \sum_{i=1}^{n} y^i / n$.
Compute the mean objective ranking: $\bar{r} \leftarrow (1 + n)/2$.
**for** $i = 1$ **to** $n$ **do**
    **for each** $S \in \mathbb{S}_x^i$ **do**
        $f_{\mathrm{obm}}(S) \leftarrow f_{\mathrm{obm}}(S) + 1/y^i$.
        $f_{\mathrm{rbm}}(S) \leftarrow f_{\mathrm{rbm}}(S) + 1/r^i$.
        $\bar{x}_S \leftarrow \bar{x}_S + 1/n$.
        $\sigma_S^{xy} \leftarrow \sigma_S^{xy} + (y^i - \bar{y})$.
        $\sigma_S^{xr} \leftarrow \sigma_S^{xr} + (r^i - \bar{r})$.
Compute the objective variance: $\sigma_y \leftarrow \sum_{i=1}^{n} (y^i - \bar{y})^2$.
Compute the ranking variance: $\sigma_r \leftarrow n(n+1)(n-1)/12$.
**for each** $S \in \tilde{\mathbb{S}}$ **do**
    $\sigma_S^x \leftarrow \bar{x}_S(1 - \bar{x}_S)n$.
    $f_{12}(S) \leftarrow \sigma_S^{xy}/\sqrt{\sigma_S^x \sigma_y}$.
    $f_{13}(S) \leftarrow \sigma_S^{xr}/\sqrt{\sigma_S^x \sigma_r}$.

---

**Lemma A.3.** *For a binary variable $x_S^i$ and any variable $y^i$, the following equality holds:*

$$\sigma_S^{xy} = \sum_{i=1}^{n} (x_S^i - \bar{x}_S)(y^i - \bar{y}) = \sum_{i \in \{1, \cdots, n\} \wedge x_S^i = 1} (y^i - \bar{y}), \tag{17}$$

*where $\bar{x}_S = \sum_{i=1}^{n} x_S^i / n$ and $\bar{y} = \sum_{i=1}^{n} y^i / n$.*

*Proof.* By making use of the fact that $\sum_{i=1}^{n} (y^i - \bar{y}) = 0$, we have

$$\sigma_S^{xy} = \sum_{i=1}^{n} (x_S^i - \bar{x}_S)(y^i - \bar{y}) = \sum_{i=1}^{n} x_S^i (y^i - \bar{y}) - \bar{x}_S \sum_{i=1}^{n} (y^i - \bar{y}) = \sum_{i=1}^{n} x_S^i (y^i - \bar{y}). \tag{18}$$

As $x_S^i$ is a binary variable,

$$\sigma_S^{xy} = \sum_{i \in \{1, \cdots, n\} \wedge x_S^i = 1} 1 \cdot (y^i - \bar{y}) + \sum_{i \in \{1, \cdots, n\} \wedge x_S^i = 0} 0 \cdot (y^i - \bar{y}) = \sum_{i \in \{1, \cdots, n\} \wedge x_S^i = 1} (y^i - \bar{y}). \tag{19}$$

$\square$

Lemma A.3 holds for any variable $y^i$. By replacing $y^i$ with $r^i$ in Lemma A.3, the convariance between the values of $x_S$ and the ranking of objective values $r$ can be computed as

$$\sigma_S^{xr} = \sum_{i=1}^{n} (x_S^i - \bar{x}_S)(r^i - \bar{r}) = \sum_{i \in \{1, \cdots, n\} \wedge x_S^i = 1} (r^i - \bar{r}). \tag{20}$$

Furthermore, as the ranking of the objective values $r$ is from 1 to $n$, it is not difficult to show that $\bar{r} = \sum_{i=1}^{n} r^i / n = (n+1)/2$ and $\sigma_r = \sum_{i=1}^{n} (r^i - \bar{r})^2 = n(n+1)(n-1)/12$.

With this simplification, the statistical features can be computed efficiently using the set representation of sample solutions $\{\mathbb{S}_x^1, \cdots, \mathbb{S}_x^n\}$. The idea is to iterate through each $S$ in the sample solutions $\mathbb{S}_x^i$ (where $i = 1, 2, \cdots, n$) and accumulate $f_{\mathrm{obm}}(S)$, $f_{\mathrm{rbm}}(S)$, $\bar{x}_S$, $\sigma_S^{xy}$ and $\sigma_S^{xr}$. The two correlation-based features can then be easily computed, as shown in Algorithm A.2. The time complexity of computing the statistical features using Algorithm A.2 is $\mathcal{O}(|\tilde{\mathbb{S}}| + \sum_{i=1}^{n} |\mathbb{S}_x^i|)$, which is much smaller than $\mathcal{O}(n|\tilde{\mathbb{S}}|)$, because $|\mathbb{S}_x^i|$ is much smaller than $|\tilde{\mathbb{S}}|$ in general.

---

**Algorithm A.3** Computing Optimal Solutions for Labeling

---

**Input:** a graph $G(V, E)$, the full set of MISs $\mathbb{S}$.
**Output:** optimal solution value $x_S^*$ for each $S \in \mathbb{S}$.
$\boldsymbol{x}^*, y^* \leftarrow Gurobi(G, \mathbb{S})$.
**for each** $S \in \mathbb{S}$ **do**
    **if** $x_S^* = 0$ **then**
        $\hat{\boldsymbol{x}}, \hat{y} \leftarrow Gurobi(G, \mathbb{S}, x_S \leftarrow 1)$.
        **if** $\hat{y} = y^*$ **then**
            **for each** $S \in \mathbb{S}$ **do**
                **if** $\hat{x}_S = 1$ **then**
                    $x_S^* \leftarrow 1$.

---

### A.3 COMPUTING CLASS LABELS

We present in Algorithm A.3 a "brute-force" approach to compute for each MIS in a graph whether it belongs to *any* optimal solution. The algorithm starts by solving a given problem instance to optimality using the exact solver Gurobi, with an optimal solution $\mathbf{x}^*$ and the optimal objective value $y^*$ obtained. The algorithm then iterates through each $S \in \mathbb{S}$. If $S$ does not belong to any optimal solution found so far, $x_S$ is fixed to 1, and the corresponding subproblem is solved by Gurobi. If the best objective value found ($\hat{y}$) is equal to $y^*$, it means that there exists at least one alternative optimal solution that includes $S$; hence all the MISs used in the newly generated optimal solution (denoted as $\hat{\boldsymbol{x}}$) are then marked as optimal.

This approach may not be the most efficient method to compute the class label for the MISs in a graph. There might exist better ways to compute all optimal solutions of a problem instance, e.g., after an optimal solution is found, a cut can be generated to prune the optimal solution from the search space, so that a different optimal solution is generated one at a time until all have been enumerated. However, as training is conducted offline, Algorithm A.3 is sufficient to construct a training set.

## B SUPPLEMENTARY EXPERIMENTS

### B.1 VISUALIZING TRAINING SET

To visualize the distribution of the training instances (i.e., MIS), we create a 2-D feature space via principal component analysis in Figure B.1. We can observe that the positive and negative training instances are separated to some extent. Most of the negative training instances are located in the left region, while the positive training instances are mainly in the middle of the reduced feature space.

### B.2 FEATURE IMPORTANCE

To investigate the relevance of the features, we compute the Pearson correlation coefficient between each feature and the class label, which is shown in Table B.1. The reduced cost ($f_{15}$) has the highest correlation with the class label, indicating it is one of the most important features. The size of a MIS ($f_1$) and the maximum, minimum and average relative sizes of a MIS ($f_2$ to $f_4$) have a noticeable correlation with the class label. As expected, the objective-based measure ($f_{10}$) and the ranking-based measure ($f_{11}$) are positively correlated with the class label, whilst the two correlation-based measures ($f_{12}$ and $f_{13}$) are negatively correlated with the class label.

We also compute the mutual information between each feature and the class label to capture any non-linear relationships between them. Since mutual information can be only computed for discrete variables, we discretize the values of each feature into five bins. The mutual information is normalized by the minimum entropy of the corresponding feature and the class label, so that it is in the range of 0 and 1. The LP features ($f_{14}$ and $f_{15}$) have the highest nonlinear correlation with the class label, as shown in Table B.1.

In addition, we present in Table B.1 the optimal weights ($\boldsymbol{w}$) of the features learned by the linear SVM classifier, which can be directly used to evaluate the quality of MISs.

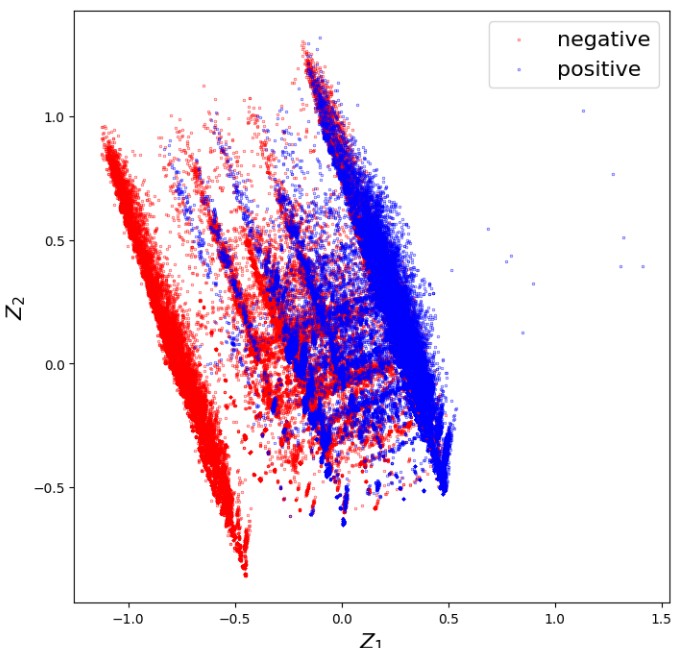

Figure B.1: The distribution of training instances in the reduced feature space, where $Z_1$ and $Z_2$ are the first two principal components of the feature vectors. Each dot represents a training instance.

Table B.1: The Pearson correlation coefficient (PCC) and normalized mutual information (NMI) between each feature and the class label, and the weight learned by linear SVM for each feature.

| Feature | Name | PCC | NMI | weight |
|---|---|---|---|---|
| $f_1$ | size | 0.2402 | 0.0389 | −0.0169 |
| $f_2$ | max relative size | 0.2417 | 0.0441 | 1.1620 |
| $f_3$ | min relative size | 0.2524 | 0.0450 | −0.9010 |
| $f_4$ | ave relative size | 0.2726 | 0.0470 | 0.2562 |
| $f_5$ | std relative size | −0.0494 | 0.0011 | −2.7004 |
| $f_6$ | max degree | −0.0249 | 0.0135 | −2.2259 |
| $f_7$ | min degree | 0.1372 | 0.0058 | 2.7940 |
| $f_8$ | ave degree | 0.0915 | 0.0084 | −0.0505 |
| $f_9$ | std degree | −0.1369 | 0.0361 | 2.9416 |
| $f_{10}$ | obj-based measure | 0.1225 | 0.0236 | 1.6858 |
| $f_{11}$ | rank-based measure | 0.1167 | 0.0209 | −0.4886 |
| $f_{12}$ | obj correlation | −0.0915 | 0.0084 | −2.7936 |
| $f_{13}$ | rank correlation | −0.0895 | 0.0082 | 1.0901 |
| $f_{14}$ | LP solution value | 0.1086 | 0.0934 | 0.9035 |
| $f_{15}$ | reduced cost | −0.5676 | 0.2603 | −2.1902 |

Finally, we investigate the trained Decision Tree (DT) to identify which features are important. When the depth of tree is set to 5, DT achieves an accuracy of 80% with all features used except the min relative size ($f_3$), obj correlation ($f_{12}$) and rank correlation ($f_{13}$). When the depth of tree is set to 3, DT achieves an accuracy of 78% with three features used: reduced cost ($f_{15}$), max degree ($f_6$) and min degree ($f_7$). Hence, the reduced cost ($f_{15}$), max degree ($f_6$) and min degree ($f_7$) are the most important features identified by DT. The trained DT with depth = 3 is presented in Figure B.2.

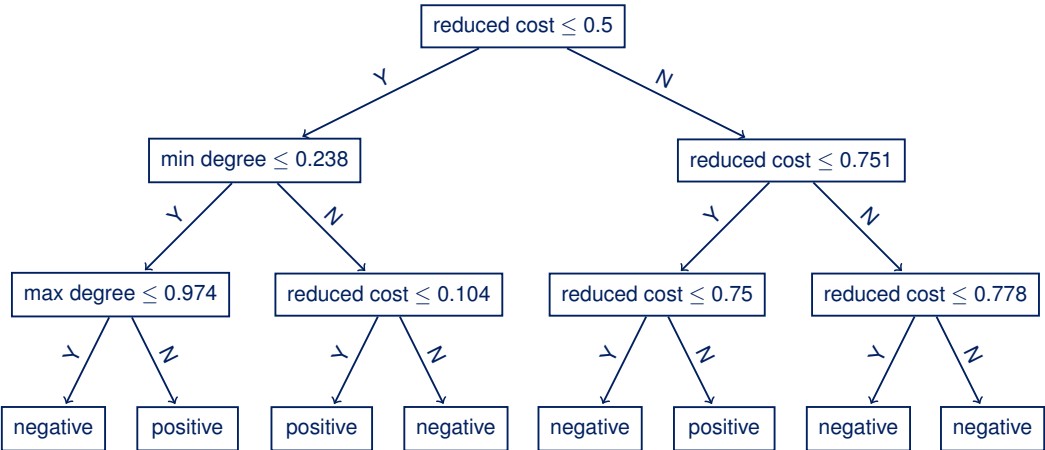

Figure B.2: The trained decision tree (with depth = 3) for predicting which MISs belonging to the optimal solutions.

Table B.2: The experimental results of our MLCG method when using different classifiers to evaluate the quality of columns on the test problem instances. The results are averaged across 25 independent runs. The last row presents the p-value of the $t$-tests between the results generated by linear SVM (with $C = 1$) and DT with different depths. The ones with statistical significance (p-value $< 0.05$) are highlighted in bold.

| Graph | Optimality Gap (%) | | | | | Runtime | | | | |
|---|---|---|---|---|---|---|---|---|---|---|
| | SVM | DT (d=5) | DT (d=10) | DT (d=15) | DT (d=20) | SVM | DT (d=5) | DT (d=10) | DT (d=15) | DT (d=20) |
| g134 | 10.00 | 10.00 | 45.60 | 20.00 | 20.00 | 661.32 | 744.72 | 1.49 | 34.24 | 34.69 |
| g135 | 5.00 | 8.33 | 25.67 | 16.33 | 16.33 | 93.39 | 7.65 | 1.73 | 28.08 | 11.74 |
| g144 | 10.00 | 10.00 | 11.20 | 10.00 | 10.40 | 1.48 | 1.79 | 1.92 | 2.36 | 2.17 |
| g171 | 0.00 | 0.00 | 16.00 | 10.50 | 7.00 | 1.07 | 1.07 | 1.27 | 1.48 | 1.27 |
| g173 | 0.00 | 10.50 | 7.00 | 10.00 | 12.50 | 1.21 | 1.08 | 1.79 | 1.21 | 1.06 |
| g175 | 0.00 | 0.50 | 1.00 | 9.50 | 6.50 | 1.32 | 1.66 | 1.68 | 1.36 | 1.27 |
| g214 | 0.00 | 0.00 | 0.00 | 1.14 | 1.14 | 0.89 | 0.92 | 0.86 | 0.90 | 0.95 |
| g243 | 10.00 | 10.40 | 20.00 | 20.00 | 20.00 | 41.83 | 33.00 | 29.30 | 37.69 | 42.36 |
| g263 | 0.80 | 1.60 | 0.00 | 14.40 | 16.00 | 1.15 | 1.10 | 1.03 | 1.02 | 0.94 |
| g324 | 0.00 | 2.50 | 20.50 | 12.00 | 4.50 | 1.48 | 1.13 | 1.17 | 1.28 | 2.36 |
| g329 | 0.00 | 0.00 | 3.20 | 0.80 | 1.60 | 0.98 | 0.98 | 1.12 | 1.15 | 1.53 |
| g348 | 0.00 | 0.00 | 0.00 | 0.00 | 0.57 | 0.94 | 1.08 | 1.04 | 0.89 | 1.08 |
| g417 | 11.00 | 9.50 | 12.00 | 12.50 | 12.50 | 1.57 | 1.29 | 1.81 | 1.47 | 2.11 |
| g420 | 0.00 | 0.00 | 4.57 | 0.00 | 0.00 | 1.07 | 1.12 | 1.11 | 0.88 | 0.95 |
| g474 | 11.11 | 11.56 | 11.56 | 20.00 | 21.33 | 1.81 | 3.73 | 4.71 | 1.76 | 2.33 |
| g479 | 0.00 | 0.00 | 0.00 | 0.00 | 0.67 | 1.08 | 1.07 | 1.13 | 1.04 | 1.60 |
| g486 | 0.00 | 3.24 | 8.38 | 4.76 | 4.76 | 15.45 | 13.12 | 1.37 | 2.52 | 3.72 |
| g487 | 0.00 | 0.00 | 4.31 | 7.69 | 7.69 | 30.89 | 19.69 | 15.42 | 9.95 | 17.45 |
| g493 | 0.00 | 35.33 | 14.67 | 10.67 | 14.00 | 1.42 | 0.92 | 1.09 | 1.31 | 1.10 |
| g496 | 8.00 | 0.00 | 0.57 | 0.00 | 4.00 | 0.99 | 0.96 | 1.11 | 1.12 | 1.15 |
| g498 | 0.00 | 0.00 | 0.00 | 4.57 | 4.57 | 0.90 | 0.85 | 0.97 | 0.85 | 0.90 |
| g508 | 0.00 | 10.67 | 8.00 | 2.22 | 0.00 | 1.14 | 1.42 | 0.89 | 0.82 | 0.88 |
| g509 | 9.78 | 17.33 | 23.11 | 17.78 | 13.33 | 1.12 | 1.01 | 1.04 | 0.85 | 1.08 |
| g511 | 0.00 | 12.89 | 12.00 | 10.22 | 0.89 | 0.87 | 0.90 | 0.90 | 0.81 | 0.93 |
| g530 | 0.00 | 0.63 | 19.58 | 12.00 | 9.05 | 1.53 | 1.67 | 0.93 | 1.35 | 1.25 |
| g556 | 0.00 | 3.03 | 9.09 | 7.64 | 9.09 | 350.30 | 325.36 | 9.36 | 43.85 | 46.08 |
| g557 | 0.00 | 0.00 | 3.06 | 0.00 | 5.88 | 26.26 | 17.71 | 4.70 | 8.93 | 3.21 |
| g597 | 0.00 | 0.00 | 18.00 | 6.43 | 7.14 | 86.31 | 53.01 | 1.62 | 112.95 | 46.83 |
| g598 | 0.29 | 0.60 | 15.14 | 7.14 | 8.00 | 133.16 | 90.77 | 1.79 | 8.09 | 8.74 |
| g617 | 0.00 | 0.00 | 0.50 | 0.50 | 0.00 | 0.98 | 0.84 | 0.98 | 0.92 | 0.97 |
| g876 | 5.14 | 25.71 | 17.14 | 14.29 | 14.86 | 1.32 | 1.08 | 1.08 | 1.21 | 1.13 |
| g894 | 0.00 | 1.14 | 0.00 | 0.00 | 0.00 | 1.06 | 1.06 | 1.12 | 1.12 | 1.52 |
| g913 | 1.14 | 20.57 | 14.29 | 14.29 | 8.00 | 1.13 | 0.88 | 1.10 | 0.90 | 1.48 |
| Mean | 2.49 | 6.24 | 10.49 | 8.41 | 7.95 | 44.47 | 40.44 | 2.99 | 9.53 | 7.48 |
| P-value | - | **1.28e-02** | **1.22e-05** | **3.01e-07** | **2.31e-07** | - | 3.31e-01 | 7.30e-02 | 1.06e-01 | 8.73e-02 |

## B.3 COMPARISON BETWEEN DIFFERENT CLASSIFIERS

We test two efficient classification algorithms, linear SVM and DT, for evaluating the quality of MISs, and the results are presented in Table B.2. *The optimality gap is computed based on the lower*

Table B.3: The experimental results of our MLCG method when using different methods to generate columns on the test problem instances. The results are averaged across 25 independent runs. The last row presents the p-value of the $t$-tests between the results generated by the two methods.

| Graph | Optimality Gap (%) | | Runtime | |
|---|---|---|---|---|
| | Random | Crossover | Random | Crossover |
| g134 | 10.00 | 10.00 | 661.32 | 268.08 |
| g135 | 5.00 | 4.00 | 93.39 | 105.96 |
| g144 | 10.00 | 10.00 | 1.48 | 1.57 |
| g171 | 0.00 | 0.00 | 1.07 | 1.28 |
| g173 | 0.00 | 0.00 | 1.21 | 1.33 |
| g175 | 0.00 | 0.00 | 1.32 | 1.16 |
| g214 | 0.00 | 0.00 | 0.89 | 0.83 |
| g243 | 10.00 | 10.00 | 41.83 | 28.37 |
| g263 | 0.80 | 5.60 | 1.15 | 1.08 |
| g324 | 0.00 | 0.00 | 1.48 | 1.18 |
| g329 | 0.00 | 0.00 | 0.98 | 1.00 |
| g348 | 0.00 | 3.43 | 0.94 | 0.95 |
| g417 | 11.00 | 10.00 | 1.57 | 1.35 |
| g420 | 0.00 | 0.00 | 1.07 | 1.06 |
| g474 | 11.11 | 11.11 | 1.81 | 3.84 |
| g479 | 0.00 | 0.00 | 1.08 | 1.03 |
| g486 | 0.00 | 0.00 | 15.45 | 17.66 |
| g487 | 0.00 | 0.00 | 30.89 | 36.33 |
| g493 | 0.00 | 0.00 | 1.42 | 1.21 |
| g496 | 8.00 | 5.14 | 0.99 | 0.81 |
| g498 | 0.00 | 2.29 | 0.90 | 0.73 |
| g508 | 0.00 | 0.44 | 1.14 | 0.82 |
| g509 | 9.78 | 11.11 | 1.12 | 0.90 |
| g511 | 0.00 | 0.89 | 0.87 | 0.85 |
| g530 | 0.00 | 0.00 | 1.53 | 1.71 |
| g556 | 0.00 | 0.00 | 350.30 | 294.19 |
| g557 | 0.00 | 0.00 | 26.26 | 28.94 |
| g597 | 0.00 | 0.00 | 86.31 | 80.82 |
| g598 | 0.29 | 0.29 | 133.16 | 141.79 |
| g617 | 0.00 | 0.00 | 0.98 | 0.90 |
| g876 | 5.14 | 3.43 | 1.32 | 1.11 |
| g894 | 0.00 | 0.00 | 1.06 | 0.92 |
| g913 | 1.14 | 0.57 | 1.13 | 0.92 |
| Mean | 2.49 | 2.68 | 44.47 | 31.23 |
| P-value | - | 4.29e-01 | - | 2.79e-01 |

*bound generated by Gurobi with 4 CPUs and a one-hour cutoff time.* We can clearly see that our method using linear SVM generates a significantly better (smaller) optimality gap than that using DT, indicating linear SVM is more effective than DT in evaluating the quality of MISs. This is somewhat surprising, because the classification accuracy of DT is much higher than that of linear SVM. Further, DT with a larger depth ($d$) may perform even worse in evaluating the quality of MISs, although the classification accuracy improves. This indicates that DT with a large depth may be overfitted. The other reason for this may be that the decision function used may not the most appropriate criterion to evaluate the quality of MISs. For example, in a large decision tree, many leaf nodes have a score of 1, and DT cannot distinguish the MISs in those nodes, resulting in a poor performance. This issue can possibly be resolved by using a different criterion to evaluate the quality of MISs, such as using the "distance" from the feature vector of a MIS to the decision boundary of DT. However, as our aim is not to compare which classification algorithm performs the best in this task, a further investigation along this line is beyond the scope of this paper.

## B.4 COMPARISON BETWEEN RANDOM AND CROSSOVER UPDATE

We test two approaches (random and crossover) for generating new MISs to replace low-quality MISs in each iteration of our MLCG method. The experimental results on the test problem in-

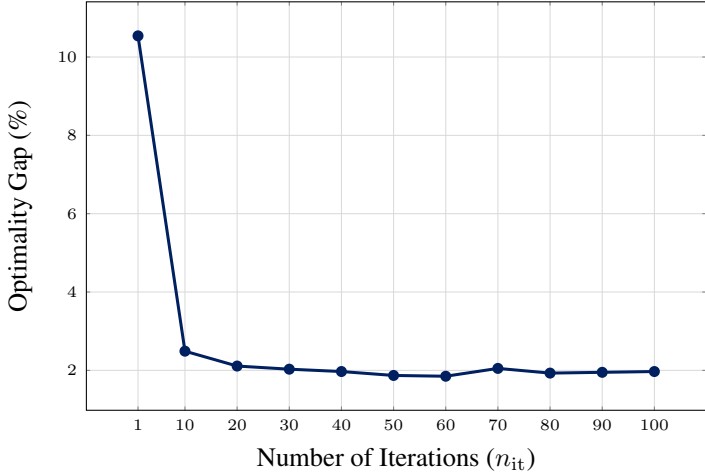

Figure B.3: The optimality gap (%) generated by our MLCG method at different iterations ($n_{it}$). The results are averaged over 25 independent runs and 33 MATILDA test problem instances.

stances are presented in Table B.3. The two approaches perform equally well (without statistically significant difference), with the average optimality gap generated by the random approach $2.49\%$ and that of the crossover approach $2.68\%$. The reason why the crossover approach is not more effective may be that the diversity of the generated MISs is important. In particular, having more high-quality but similar MISs in the subset is not expected to result in a better integer solution. In the rest of the paper, we simply use the random approach to generate new MISs.

## B.5 EFFECTS OF NUMBER OF ITERATIONS

We investigate the effect of the number of iterations ($n_{it}$) on the performance of our MLCG method. To do so, we vary the number of iterations from 1 to 100, and present the average optimality gap generated by our method in Figure B.3. We can observe that the solution quality generated by our method improves (i.e., optimality gap decreases) dramatically in the first several iterations, and the improvement slows down later on. The best (smallest) optimality gap is generated at around the $50^{th}$ iteration, and the solution quality cannot be further improved afterwards. Note that the result presented in Figure B.3 is the best optimality gap generated at each iteration, instead of the best optimality gap found so far. Therefore, the curve in Figure B.3 does not monotonically decrease due to randomness. The detailed results for each test problem instance are presented in Table B.4.

## B.6 DETAILED EXPERIMENTS FOR SEEDING RESTRICTED MASTER HEURISTIC

We use the columns generated by our MLCG method to warm-start RMH, namely MLRMH, compared to RMH with the typical random initialization. The number of initial MISs is set to $10|V|$. The restricted master problem and the pricing problem are both solved by Gurobi. The detailed experimental results for the 33 MATILDA test problem instances are presented in Table B.5. Note that the optimal LP objective is a valid lower bound (LB) for the objective value of integer solutions, and is used to compute the optimality gap in Table B.5. We can observe that the LB generated by MLRMH is very tight for most of the problem instances tested. Significantly, our MLRMH algorithm is able to prove optimality for many of the problem instances and provide an optimality gap for the remainder, making it a quasi-exact method. MLRMH consistently generates an equally-well or significantly better solution than RMH for the test problem instances except g496. The mean optimality gap generated by MLRMH is much smaller than that of RMH. Interestingly, by using the columns generated by our MLCG method to warm-start RMH, the number of CG iterations is significantly reduced from 69.63 to 7.08 on average. This also leads to a significant reduction in the runtime of solving the LP and MIP. Note that the time used by our MLCG method to generate columns is counted as part of the LP solving time.

Table B.4: The average optimality gap (%) generated by our MLCG method at different iterations on the test problem instances. The optimality gap is averaged over 25 independent runs. The last row presents the p-value of the $t$-tests between the results generated at the first iteration and each of the other iterations. The ones with statistical significance (p-value $< 0.05$) are highlighted in bold.

| Graph | Iter 1 | Iter 10 | Iter 20 | Iter 30 | Iter 40 | Iter 50 | Iter 60 | Iter 70 | Iter 80 | Iter 90 | Iter 100 |
|---|---|---|---|---|---|---|---|---|---|---|---|
| g134 | 22.40 | 10.00 | 10.00 | 10.00 | 10.00 | 10.00 | 10.00 | 10.00 | 10.00 | 10.00 | 10.00 |
| g135 | 13.00 | 5.00 | 1.33 | 0.00 | 0.00 | 0.67 | 0.33 | 0.33 | 1.00 | 0.67 | 1.00 |
| g144 | 12.40 | 10.00 | 10.00 | 10.00 | 10.00 | 10.00 | 10.00 | 10.00 | 10.00 | 10.00 | 10.00 |
| g171 | 11.50 | 0.00 | 0.00 | 0.00 | 0.00 | 0.00 | 0.00 | 0.00 | 0.00 | 0.00 | 0.00 |
| g173 | 12.50 | 0.00 | 0.00 | 0.00 | 0.00 | 0.00 | 0.00 | 0.00 | 0.00 | 0.00 | 0.00 |
| g175 | 12.00 | 0.00 | 0.00 | 0.00 | 0.00 | 0.00 | 0.00 | 0.00 | 0.00 | 0.00 | 0.00 |
| g214 | 12.00 | 0.00 | 0.00 | 0.00 | 0.00 | 0.00 | 0.00 | 0.00 | 0.00 | 0.00 | 0.00 |
| g243 | 20.00 | 10.00 | 10.00 | 10.00 | 10.00 | 10.00 | 10.00 | 10.00 | 10.00 | 10.00 | 10.00 |
| g263 | 17.60 | 0.80 | 0.00 | 0.00 | 0.00 | 0.00 | 0.00 | 0.00 | 0.00 | 0.00 | 0.00 |
| g324 | 5.00 | 0.00 | 0.00 | 0.00 | 0.00 | 0.00 | 0.00 | 0.00 | 0.00 | 0.00 | 0.00 |
| g329 | 6.40 | 0.00 | 0.00 | 0.00 | 0.00 | 0.00 | 0.00 | 0.00 | 0.00 | 0.00 | 0.00 |
| g348 | 10.29 | 0.00 | 0.00 | 0.00 | 0.00 | 0.00 | 0.00 | 0.00 | 0.00 | 0.00 | 0.00 |
| g417 | 24.00 | 11.00 | 8.00 | 7.00 | 5.50 | 2.00 | 3.00 | 2.50 | 2.00 | 4.00 | 3.50 |
| g420 | 0.00 | 0.00 | 0.00 | 0.00 | 0.00 | 0.00 | 0.00 | 0.00 | 0.00 | 0.00 | 0.00 |
| g474 | 22.22 | 11.11 | 11.11 | 11.11 | 11.11 | 11.11 | 11.11 | 11.11 | 11.11 | 11.11 | 11.11 |
| g479 | 6.67 | 0.00 | 0.00 | 0.00 | 0.00 | 0.00 | 0.00 | 0.00 | 0.00 | 0.00 | 0.00 |
| g486 | 4.76 | 0.00 | 0.00 | 0.19 | 0.00 | 0.19 | 0.00 | 0.19 | 0.00 | 0.00 | 0.00 |
| g487 | 8.92 | 0.00 | 0.00 | 0.00 | 0.00 | 0.00 | 0.00 | 0.00 | 0.00 | 0.00 | 0.00 |
| g493 | 10.67 | 0.00 | 0.00 | 0.00 | 0.00 | 0.00 | 0.00 | 0.00 | 0.00 | 0.00 | 0.00 |
| g496 | 3.43 | 8.00 | 6.29 | 8.00 | 6.29 | 7.43 | 5.71 | 11.43 | 6.29 | 9.14 | 6.29 |
| g498 | 12.00 | 0.00 | 0.00 | 0.00 | 0.00 | 0.00 | 0.57 | 0.00 | 1.14 | 0.00 | 0.00 |
| g508 | 0.00 | 0.00 | 0.00 | 0.00 | 0.00 | 0.00 | 0.00 | 0.00 | 0.00 | 0.00 | 0.00 |
| g509 | 10.67 | 9.78 | 10.67 | 10.22 | 10.67 | 8.44 | 10.22 | 9.33 | 9.78 | 9.33 | 10.67 |
| g511 | 0.00 | 0.00 | 0.00 | 0.00 | 0.00 | 0.00 | 0.00 | 0.00 | 0.00 | 0.00 | 0.00 |
| g530 | 8.84 | 0.00 | 0.00 | 0.00 | 0.00 | 0.00 | 0.00 | 0.00 | 0.00 | 0.00 | 0.00 |
| g556 | 14.55 | 0.00 | 0.00 | 0.00 | 0.00 | 0.00 | 0.00 | 0.00 | 0.00 | 0.00 | 0.00 |
| g557 | 5.65 | 0.00 | 0.00 | 0.00 | 0.00 | 0.00 | 0.00 | 0.00 | 0.00 | 0.00 | 0.00 |
| g597 | 7.14 | 0.00 | 0.00 | 0.00 | 0.00 | 0.00 | 0.00 | 0.00 | 0.00 | 0.00 | 0.00 |
| g598 | 14.00 | 0.29 | 0.00 | 0.00 | 0.29 | 0.00 | 0.00 | 0.00 | 0.00 | 0.00 | 0.00 |
| g617 | 1.50 | 0.00 | 0.00 | 0.00 | 0.00 | 0.00 | 0.00 | 0.00 | 0.00 | 0.00 | 0.00 |
| g876 | 14.29 | 5.14 | 2.29 | 0.57 | 1.14 | 1.71 | 0.00 | 1.14 | 2.29 | 0.00 | 2.29 |
| g894 | 9.14 | 0.00 | 0.00 | 0.00 | 0.00 | 0.00 | 0.00 | 0.00 | 0.00 | 0.00 | 0.00 |
| g913 | 14.29 | 1.14 | 0.00 | 0.00 | 0.00 | 0.00 | 0.00 | 1.71 | 0.00 | 0.00 | 0.00 |
| Mean | 10.54 | 2.49 | 2.11 | 2.03 | 1.97 | 1.87 | 1.85 | 2.05 | 1.93 | 1.95 | 1.97 |
| P-value | - | **2.59e-10** | **2.12e-10** | **4.42e-10** | **3.04e-10** | **5.14e-10** | **3.65e-10** | **2.51e-09** | **4.68e-10** | **8.27e-10** | **4.26e-10** |

So far, we have only used the 33 MATILDA problem instances from (Smith-Miles & Bowly, 2015) as our test problem instances. The MISs in those graphs can be enumerated using 16GB memory. Here, we include 30 other MATILDA problem instances from (Smith-Miles & Bowly, 2015), in which the number of MISs is so large that they cannot be enumerated with 16GB memory. We evaluate the efficacy of our MLRMH method on these problem instances. Table B.6 presents the experimental results of our MLRMH algorithm compared against the RMH algorithm for solving the 30 MATILDA test problem instances. Our MLRMH algorithm is able to prove optimality for 11 problem instances, while the RMH algorithm is only able to prove optimality for 8 instances. Further, the average optimality gap generated by our MLRMH method (8.96%) is statistically significantly better than that generated by the RMH algorithm (12.53%). By using the columns generated by our method to warm-start RMH, the average number of CG iterations is significantly reduced from 156.28 to 59.38. This also leads to a significant reduction in the runtime of solving the LP and MIP. This clearly demonstrates that the RMH algorithm using the columns generated by our method as a warm start performs significantly better than the RMH alone.

Finally, we use a set of *larger* DIMACS graphs, the number of vertices in which varies from 125 to 1,000, to evaluate the generalization capability of our ML model. The cutoff time of MLRMH or RMH for solving each problem instance is set to ten hours. Table B.7 presents the experimental results (averaged over two independent runs) of the problem instances, whose LP relaxation can be optimally solved via CG within the cutoff time. Note that Gurobi with a ten-hour cutoff time may not be able to solve the reduced MIPs for large problem instances (e.g., r1000.1), resulting in a large optimality gap. Further, the time used by our MLCG method in generating columns for large problem instances may be non-negligible, because the computation of features requires solving the restricted master problem in each iteration of our algorithm. However, overall the RMH algorithm using the columns generated by our MLCG method as a warm start performs better than RMH alone.

Table B.5: The experimental results of MLRMH and RMH for solving the 33 MATILDA test problem instances. The last row presents the p-value of the $t$-tests between the results generated by MLRMH and RMH. The ones with statistical significance (p-value $< 0.05$) are in bold.

| Graph | Density | LB | Optimality Gap (%) | | CG Iterations | | LP Runtime | | Total Runtime | |
|---|---|---|---|---|---|---|---|---|---|---|
| | | | MLRMH | RMH | MLRMH | RMH | MLRMH | RMH | MLRMH | RMH |
| g134 | 0.34 | 10 | 10.00 | 16.40 | 28.52 | 206.88 | 45.55 | 258.56 | 236.12 | 544.03 |
| g135 | 0.33 | 12 | 0.00 | 8.33 | 1.00 | 31.12 | 5.60 | 11.96 | 42.99 | 39.68 |
| g144 | 0.37 | 10 | 10.00 | 10.00 | 1.00 | 18.36 | 4.21 | 4.04 | 4.72 | 5.92 |
| g171 | 0.28 | 7 | 14.29 | 14.29 | 34.56 | 196.28 | 9.87 | 35.41 | 10.01 | 35.74 |
| g173 | 0.36 | 8 | 0.00 | 7.00 | 4.92 | 110.28 | 4.59 | 15.37 | 4.81 | 16.67 |
| g175 | 0.32 | 8 | 0.00 | 10.50 | 3.24 | 14.24 | 4.15 | 1.49 | 4.46 | 2.21 |
| g214 | 0.26 | 7 | 0.00 | 2.29 | 1.04 | 22.56 | 3.96 | 1.84 | 4.06 | 2.14 |
| g243 | 0.30 | 10 | 10.00 | 20.00 | 3.04 | 96.60 | 7.58 | 46.05 | 25.36 | 60.95 |
| g263 | 0.23 | 5 | 0.00 | 17.60 | 1.00 | 1.56 | 4.79 | 0.08 | 5.13 | 0.35 |
| g324 | 0.38 | 8 | 0.00 | 0.00 | 7.52 | 31.16 | 5.25 | 4.14 | 5.47 | 5.05 |
| g329 | 0.21 | 5 | 0.00 | 7.20 | 1.00 | 1.00 | 4.71 | 0.05 | 4.88 | 0.33 |
| g348 | 0.21 | 7 | 0.00 | 0.00 | 11.40 | 98.04 | 4.57 | 7.36 | 4.65 | 7.48 |
| g417 | 0.40 | 8 | 3.50 | 12.00 | 22.36 | 254.72 | 14.86 | 159.33 | 15.39 | 159.68 |
| g420 | 0.41 | 6 | 16.67 | 16.67 | 2.28 | 55.24 | 4.39 | 13.96 | 4.53 | 14.36 |
| g474 | 0.34 | 9 | 11.11 | 20.00 | 1.00 | 25.76 | 5.07 | 6.20 | 6.35 | 7.86 |
| g479 | 0.23 | 6 | 0.00 | 0.00 | 11.08 | 59.12 | 5.06 | 6.26 | 5.22 | 6.63 |
| g486 | 0.67 | 21 | 0.00 | 4.57 | 1.04 | 44.08 | 6.79 | 53.19 | 28.63 | 58.79 |
| g487 | 0.41 | 12 | 8.33 | 8.33 | 25.24 | 180.68 | 45.98 | 165.59 | 69.10 | 196.42 |
| g493 | 0.24 | 6 | 0.00 | 3.33 | 1.04 | 8.80 | 5.11 | 0.79 | 5.35 | 1.18 |
| g496 | 0.30 | 7 | 2.29 | 0.00 | 14.88 | 11.16 | 4.64 | 0.79 | 4.92 | 1.03 |
| g498 | 0.35 | 7 | 0.00 | 2.86 | 1.00 | 4.00 | 4.51 | 0.22 | 4.69 | 0.36 |
| g508 | 0.53 | 9 | 0.00 | 0.00 | 1.00 | 1.04 | 3.56 | 0.06 | 3.63 | 0.21 |
| g509 | 0.52 | 9 | 8.89 | 10.22 | 1.00 | 1.08 | 3.42 | 0.08 | 3.54 | 0.25 |
| g511 | 0.50 | 9 | 0.00 | 0.00 | 2.44 | 3.00 | 4.15 | 0.23 | 4.27 | 0.36 |
| g530 | 0.51 | 19 | 0.00 | 2.53 | 1.00 | 79.16 | 5.96 | 49.58 | 6.24 | 50.45 |
| g556 | 0.35 | 10 | 10.00 | 16.80 | 20.68 | 203.88 | 29.53 | 247.74 | 145.90 | 436.49 |
| g557 | 0.56 | 16 | 6.25 | 6.25 | 2.00 | 91.92 | 8.61 | 169.56 | 35.38 | 201.48 |
| g597 | 0.46 | 13 | 7.69 | 8.00 | 7.60 | 136.44 | 19.83 | 182.19 | 108.05 | 385.99 |
| g598 | 0.48 | 13 | 7.69 | 15.39 | 3.04 | 145.08 | 9.02 | 173.66 | 102.27 | 188.43 |
| g617 | 0.29 | 8 | 0.00 | 0.00 | 7.52 | 59.56 | 4.82 | 5.15 | 4.88 | 5.23 |
| g876 | 0.43 | 7 | 1.71 | 14.29 | 1.00 | 7.32 | 5.41 | 0.90 | 5.79 | 1.17 |
| g894 | 0.43 | 7 | 0.00 | 1.71 | 6.68 | 93.92 | 5.24 | 14.97 | 5.39 | 16.17 |
| g913 | 0.44 | 7 | 0.00 | 14.29 | 1.48 | 3.60 | 4.40 | 0.31 | 4.53 | 0.52 |
| Mean | - | - | 3.89 | 8.21 | 7.08 | 69.63 | 9.25 | 49.61 | 28.08 | 74.35 |
| P-value | - | - | - | **2.18e-05** | - | **5.42e-06** | - | **2.60e-02** | - | **6.20e-03** |

Table B.6: The experimental results of the MLRMH and RMH algorithms for solving the 30 MATILDA test problem instances, in which the number of MISs is so large that they cannot be enumerated upfront using 16GB memory. The last row presents the p-values of the $t$-tests.

| Graph | LB | Optimality Gap (%) | | CG Iterations | | LP Runtime | | Total Runtime | |
|---|---|---|---|---|---|---|---|---|---|
| | | MLRMH | RMH | MLRMH | RMH | MLRMH | RMH | MLRMH | RMH |
| g115 | 25 | 0.00 | 0.00 | 1.20 | 52.84 | 6.35 | 55.32 | 6.44 | 55.46 |
| g133 | 6 | 23.33 | 32.00 | 178.32 | 348.72 | 162.23 | 244.24 | 306.22 | 276.87 |
| g136 | 5 | 20.00 | 20.00 | 227.00 | 365.52 | 74.26 | 168.27 | 90.59 | 239.98 |
| g140 | 8 | 12.50 | 12.50 | 73.04 | 277.60 | 25.51 | 87.73 | 27.92 | 92.32 |
| g169 | 6 | 16.67 | 16.67 | 229.40 | 407.84 | 92.99 | 151.73 | 103.00 | 180.35 |
| g245 | 9 | 0.00 | 10.22 | 1.48 | 80.36 | 5.49 | 14.79 | 6.65 | 17.15 |
| g249 | 3 | 33.33 | 37.33 | 34.80 | 155.16 | 9.71 | 16.34 | 14.87 | 41.23 |
| g256 | 8 | 12.50 | 12.50 | 156.04 | 314.60 | 131.82 | 236.40 | 157.90 | 250.02 |
| g296 | 7 | 14.29 | 14.29 | 175.24 | 337.32 | 169.35 | 269.26 | 252.70 | 286.23 |
| g323 | 7 | 25.71 | 28.57 | 137.08 | 328.36 | 156.79 | 248.59 | 321.03 | 271.67 |
| g326 | 9 | 7.11 | 11.11 | 120.84 | 307.28 | 66.12 | 119.24 | 85.43 | 124.04 |
| g327 | 6 | 16.67 | 32.67 | 1.88 | 78.64 | 5.18 | 16.49 | 6.86 | 26.44 |
| g347 | 6 | 2.00 | 16.67 | 1.00 | 19.04 | 5.55 | 3.06 | 5.88 | 3.84 |
| g407 | 7 | 0.00 | 0.00 | 1.52 | 1.20 | 4.47 | 0.07 | 4.59 | 0.20 |
| g416 | 21 | 0.00 | 0.00 | 1.52 | 13.88 | 5.55 | 7.72 | 5.74 | 7.99 |
| g457 | 5 | 20.00 | 20.00 | 310.80 | 471.80 | 95.59 | 130.67 | 100.79 | 136.83 |
| g460 | 5 | 20.00 | 20.00 | 67.88 | 284.68 | 15.99 | 53.69 | 16.28 | 54.33 |
| g472 | 12 | 8.33 | 16.00 | 7.20 | 155.80 | 17.78 | 195.43 | 155.58 | 250.68 |
| g475 | 6 | 9.33 | 16.67 | 1.00 | 4.56 | 4.44 | 0.36 | 4.76 | 0.60 |
| g476 | 6 | 13.33 | 16.67 | 1.16 | 53.56 | 4.59 | 8.70 | 10.53 | 10.34 |
| g477 | 8 | 0.00 | 5.00 | 27.04 | 171.84 | 11.09 | 39.85 | 11.67 | 43.84 |
| g488 | 8 | 0.00 | 10.00 | 5.36 | 127.32 | 6.97 | 34.05 | 8.22 | 35.89 |
| g490 | 6 | 1.33 | 14.67 | 1.00 | 2.76 | 5.33 | 0.16 | 5.78 | 0.63 |
| g494 | 22 | 4.55 | 4.55 | 1.00 | 1.92 | 4.20 | 0.45 | 4.30 | 0.59 |
| g499 | 7 | 0.00 | 0.00 | 1.00 | 1.00 | 4.42 | 0.05 | 4.55 | 0.15 |
| g600 | 24 | 0.00 | 0.00 | 1.08 | 28.20 | 6.88 | 36.31 | 7.11 | 36.70 |
| g623 | 14 | 0.00 | 0.00 | 3.08 | 53.32 | 5.46 | 17.32 | 5.53 | 17.48 |
| g625 | 13 | 7.69 | 7.69 | 1.68 | 54.20 | 5.89 | 19.65 | 6.11 | 20.00 |
| g647 | 14 | 0.00 | 0.00 | 1.84 | 80.24 | 5.31 | 43.87 | 5.50 | 44.28 |
| g883 | 7 | 0.00 | 0.00 | 9.84 | 108.72 | 6.35 | 23.17 | 6.49 | 23.65 |
| Mean | - | 8.96 | 12.53 | 59.38 | 156.28 | 37.39 | 74.77 | 58.30 | 84.99 |
| P-value | - | - | **5.49e-04** | - | **3.53e-08** | - | **5.58e-05** | - | **9.46e-04** |

Table B.7: The experimental results of the MLRMH and RMH algorithms for solving a set of larger DIMACS problem instances. The last row presents the p-values of the $t$-tests.

| Graph | $|V|$ | LB | Optimality Gap (%) | | CG Iterations | | LP Runtime | | Total Runtime | |
|---|---|---|---|---|---|---|---|---|---|---|
| | | | MLRMH | RMH | MLRMH | RMH | MLRMH | RMH | MLRMH | RMH |
| dsjc125.1 | 125 | 5 | 20.00 | 20.00 | 589.50 | 794.00 | 507.17 | 609.36 | 680.51 | 910.71 |
| dsjc125.5 | 125 | 16 | 12.50 | 12.50 | 6.50 | 188.00 | 43.09 | 690.24 | 4363.00 | 2618.90 |
| dsjc125.9 | 125 | 43 | 2.33 | 2.33 | 1.00 | 5.00 | 15.55 | 40.52 | 15.73 | 40.88 |
| r125.1 | 125 | 5 | 20.00 | 20.00 | 1.00 | 1.00 | 6.62 | 0.09 | 29.61 | 27.95 |
| r125.1c | 125 | 46 | 0.00 | 0.00 | 1.00 | 1.00 | 5.83 | 0.73 | 5.87 | 0.78 |
| r125.5 | 125 | 36 | 0.00 | 2.78 | 1.00 | 8.50 | 7.72 | 7.46 | 7.86 | 7.71 |
| r250.1c | 250 | 64 | 0.00 | 0.00 | 1.00 | 1.00 | 27.90 | 9.33 | 27.94 | 9.49 |
| dsjc250.9 | 250 | 71 | 1.41 | 1.41 | 1.00 | 71.50 | 102.12 | 4304.90 | 119.71 | 4378.50 |
| le450_15a | 450 | 15 | 66.67 | 73.33 | 207.00 | 510.00 | 571.17 | 1091.20 | 36005.00 | 36006.00 |
| le450_15b | 450 | 15 | 63.33 | 70.00 | 199.50 | 437.00 | 483.72 | 1112.40 | 36004.00 | 36008.00 |
| le450_25b | 450 | 25 | 26.00 | 32.00 | 1.00 | 2.00 | 104.13 | 3.08 | 36003.00 | 36009.00 |
| r1000.1 | 1000 | 20 | 82.50 | 90.00 | 1.00 | 3.50 | 505.08 | 11.46 | 36002.00 | 36002.00 |
| Mean | - | - | 24.56 | 27.03 | 84.21 | 168.54 | 198.34 | 656.74 | 12439.00 | 12668.00 |
| P-value | - | - | - | **2.22e-02** | - | **2.63e-02** | - | 2.21e-01 | - | 5.73e-01 |