# OpenReview forum: "Learning to Generate Columns with Application to Vertex Coloring"
_ICLR.cc/2023/Conference — ICLR 2023 poster_

### Official Review · Reviewer_igoj · 2022-10-19

**Confidence:** 4
**Clarity, Quality, Novelty And Reproducibility:** The paper is clearly written. The tec…
**Correctness:** 4
**Technical Novelty And Significance:** 2
**Empirical Novelty And Significance:** 3
**Recommendation:** 6

**Strength And Weaknesses:**

Strength:
- A motivated new approach to combinatorial optimization


Weaknesses
- The approach is not general, i.e., specifically designed for a series of optimization problems generated from similar distributions
- The idea itself is interesting, but there is no new technique introduced

The proposed idea of learning relevant columns (MISs) from a series of problems would make sense when such similar problems are available. But, the approach has the limitation that it is not suitable to apply for new problems obtained from different distributions. The learning techniques themselves are quite standard. So, I feel that the technical contribution of the paper is not significantly strong though the idea itself is promising.

**Summary Of The Paper:**

The paper proposes a machine-learning-based approach for generating columns for LP relaxations of vertex cover problems (VCPs). The paper (implicitly) assumes that the VCPs are generated from some distribution and it proposes to learn a classifier to predict if a MIS is a part of the optimal solution, given a training set of VCP problem instances. Then, the standard column generation algorithm is used with the learned classifier to choose columns. The experiments using real data sets show that the proposed approach tends to produce better solutions.

**Summary Of The Review:**

 The idea itself is promising, but the technical contribution of the paper seem not significantly strong.

---

> ### Author Response · Authors · 2022-11-18
> **Response to Reviewer igoj**
>
> Thank you for the overall positive feedback and suggestions to further improve our paper. We have carefully considered the comments and revised our manuscript accordingly (highlighted in blue).
>
> **1. The approach is not general, i.e., specifically designed for a series of optimization problems generated from similar distributions. The approach has the limitation that it is not suitable to apply for new problems obtained from different distributions.**
>
> We believe our method is not confined to solving problem instances from a similar distribution. In fact, we have used the standard benchmark instances (MATILDA and DIMACS graphs) with various problem characterises to evaluate our method in the paper.  The MATILDA graphs were evolved using Genetic Algorithm to acquire various characteristics such as density, algebraic connectivity, energy, and standard deviation of eigenvalues of adjacency matrix etc. DIMACS graphs are a collection of hard instances from different resources with very different characteristics and sizes. Our experimental results showed that our method trained on a subset of MATILDA graphs performs well across the MATILDA and DIMACS graphs. We have added the above discussion in Section 5.3 of our paper.
>
> **2. The learning techniques themselves are quite standard. The idea itself is interesting, but there is no new technique introduced.**
>
> The novelty of our paper is mainly in the modelling of column selection as a classification task and the methodology of leveraging machine learning to boost the traditional column generation method. Whilst the learning techniques used are quite standard, our method requires significantly novel designs such as feature extraction, class label computation and column sampling etc. It is particularly challenging to use the machine learning model to select columns for solving an unseen problem instance, because there exists a huge number of columns which cannot be enumerated upfront. We have developed an effective sampling and filtering method in Section 4 to deal with this challenge. Compared to other "learning to optimize" methods, our machine learning model guides the selection of useful columns which are ultimately "validated" by the column generation master problem that makes the final selection of columns. To recap, our method is a strong and novel application of machine learning in the combinatorial optimization domain.

---

### Official Review · Reviewer_795f · 2022-10-25

**Confidence:** 4
**Correctness:** 4
**Technical Novelty And Significance:** 2
**Empirical Novelty And Significance:** 2
**Recommendation:** 6

**Clarity, Quality, Novelty And Reproducibility:**

The ideas of the paper are explained very clearly. There is some novelty in the design of the features for ML algorithms, and in combining other existing algorithms.

**Strength And Weaknesses:**

Strengths:
For the specific Vertex Coloring Problem, the features that are defined are natural and prove to be relevant in characterizing the quality of MIS. The method is clear and combines well understood ML classifiers. The experimental evaluation confirms the quality of the method.

Weaknesses:
It is not clear how this method can be used for problems other than Vertex Coloring, although the authors claim that the method is generic, and mention in conclusion that vehicle routing may lend itself to a similar treatment.
The size of the test graphs is small, with only 100 vertices. Any indication of how this method would scale for larger graphs?

**Summary Of The Paper:**

The authors propose a Column Generation method for Vertex Coloring Problem, based on Machine Learning. The column in this case is a maximal independent set, which can potentially share the same color. Specific features were designed for this problem: 1) problem-specific features derived from the graph; 2) statistical measures computed from sample solutions; 3) linear program (LP) features. Various ML classifiers (KNN, Decision Trees, SVM) are trained on instances with known optimal solutions, in order to predict high quality maximal independent sets (MIS). The MLCG algorithm samples randomly generated MIS or uses a genetic algorithm like procedure to evolve new ones and keeps a subset of the highest quality ones. The generated columns are then fed into the Gurobi solver. A good sized experimental evaluation shows that MLCG outputs higher quality columns compared with other strategies, like random CG, Reduced Cost, MLF and Full.

**Summary Of The Review:**

The paper shows an improvement in Column Generating quality with the help of ML classifiers, for Vertex Coloring. While there is some novelty in the design of the features and of the method, the scope is still very focused on a single type of problem (VC), and the experimental evaluation is still limited to a small number of graphs of modest size.

---

> ### Author Response · Authors · 2022-11-18
> **Response to Reviewer 795f**
>
> Thank you for the overall positive feedback and suggestions to further improve our paper. We have carefully considered the comments and revised our manuscript accordingly (highlighted in blue).
>
> **1. It is not clear how this method can be used for problems other than Vertex Coloring, although the authors claim that the method is generic, and mention in conclusion that vehicle routing may lend itself to a similar treatment. The scope is still very focused on a single type of problem (VCP).**
>
> The methodology of using machine learning to select a subset of promising variables to reduce the problem size is generally applicable to any combinatorial optimisation problem. In this paper, we focus on the Dantzig–Wolfe reformulation of combinatorial optimisation problems, in which there are often a huge number of variables or columns. In other words, our method is particularly suitable for solving problems in which the number of variables or columns cannot be enumerated upfront. Typical examples other than vertex coloring and vehicle routing include 1) the Cutting Stock Problem in which a column is a cutting pattern generated by solving the Knapsack problem, 2) the Airline Crew Scheduling problem in which a column is a pairing determined by a constrained shortest path problem, and many more. As a proof of concept, we have demonstrated the efficacy of our method on the graph coloring problem in this paper. To extend our method to other problems such as the vehicle routing problem, it will require the design of problem specific features to characterize a column and a specialized method to generate columns. However, as the methodology is given, it should not be too difficult to implement a model for other problems. Considering the short page limit of the paper and many interesting results have to be presented in the Appendix, we would like to leave the extension of the current model to other problems as future work. We have added some of the above discussion to the conclusion section.
>
> **2. The size of the test graphs is small, with only 100 vertices. Any indication of how this method would scale for larger graphs?**
>
> We have evaluated the performance of our method on larger DIMACS graphs in which the number of vertices vary from 125 to 1000. The experimental results on summarized in Table 2, and the detailed results are presented in Table B.7 of the Appendix due to the page limit. Overall, the RMH algorithm using the columns generated by our MLCG method as a warm start performs better than RMH alone. The detailed discussion can be found in Section B.6 of the Appendix. In the revised manuscript, we have highlighted the size of DIMACS graphs in the paper.

---

### Official Review · Reviewer_DZnS · 2022-10-30

**Confidence:** 5
**Correctness:** 4
**Technical Novelty And Significance:** 3
**Empirical Novelty And Significance:** 3
**Recommendation:** 8

**Clarity, Quality, Novelty And Reproducibility:**

The clarity, quality, novelty and reproducibility of this work are all significant enough to be accepted as a conference preceding.

**Strength And Weaknesses:**

Main strength: This paper is very nicely written with little typos and easy to follow. The numerical experiments are extensive and convincing. The main idea of this paper is novel and interesting and is extendable for many other types of combinatorial optimization problems, it provides another bridge between communities of ML and MIP.

Main weakness: 1. The constructed features for training instances seem too arbitrary, a detailed explanation of the intuition and analysis can be helpful.
2. In the numerical experiment, all the benchmark methods are variants of column generation. A more detailed comparison with other state-of-art methods for VCP should be considered (either based on tighter LP relaxations or combinatorial properties of VCP). Based on the same reason, since for many classic combinatorial optimization methods there have already been quite successful combinatorial algorithms in the literature, simply extending the same idea of this paper to tackle those other problems cannot be trusted to outperform other algorithms.

minor comments/questions: 1. In the formulation given by (1)-(3), since in the graph of VCP, every node can only be colored exactly once, should the $\geq$ in (2) be equality?
2. You mentioned that the set covering formulation (1)-(3) has a tight LP relaxation. Can you add a reference for this statement? Is it also true that, if you select a subset $\tilde{\mathbb{S}}$ of $\mathbb{S}$ and replace $\mathbb{S}$, the binary IP (1)-(3) still has a tight LP relaxation?
3. In constraint (5), $\tilde{\mathbb{S}}$ should be $\tilde{\mathbb{S}}_v$, which has not been defined until later in Section 3.1.
4. In Algorithm 1, at every iteration you keep the top $\epsilon$ percentage of MISs, here in convention $\epsilon$ is used to denote small constant. Here you should either say "top $\epsilon$ of " or use some other notation such as $\kappa$.

**Summary Of The Paper:**

This paper combines the machine learning technique within the traditional column generation method for solving combinatorial optimization problems. Specifically, the authors propose to use machine learning algorithms to predict high quality columns that belong to an optimal integer solution. Here the machine learning model is trained using training instances that correspond to columns (MISs in the case of VCP), where the features of each instance are constructed from various perspectives.

The main contribution of this paper is, it presents a nice integration of MIP technique with the prediction power of machine learning. Instead of fully replying on ML predict the optimal solution for combinatorial optimization problem, here in this paper it simply leverages the power of ML technique for column selection. This idea in alignment with many recent literatures about learning to branch, learning to cut in MIP, becomes an increasingly exciting area in both communities.


**Summary Of The Review:**

Overall, I think this paper has made significant contribution to the community of MIP or CO by leveraging the power of ML techniques. Even though this type of method is completely empirical-based and often times not the optimal method in practice, such attempt is still encouraging and in my opinion, it's not impossible to see major breakthroughs in combinatorial optimization in the future through unconventional approaches like ML. Based on the novelty and potential carried in this paper, I recommend this paper to be accepted.

---

> ### Author Response · Authors · 2022-11-18
> **Response to Reviewer DZnS**
>
> Thank you for the positive feedback and suggestions to further improve our paper. We have carefully considered the comments and revised our manuscript accordingly (highlighted in blue).
>
> **1. A detailed explanation of the intuition and analysis of the constructed features can be helpful.**
>
> We have expanded the intuition/analysis of the features. The three categories of features are designed carefully, each capturing different characteristics of a maximal independent set (MIS). The problem specific features focus on the local characteristics of a MIS such as how many vertices a MIS can cover. Our statistical features are motivated by the observation that many optimization problems have a “backbone” structure. In other words, high-quality solutions potentially share some components with the optimal solution. The goal of our statistical features is to extract the shared components from high-quality solutions. The LP features are based on the LP theory, which is widely used by state-of-the-art MIP solvers. We also investigated the relevance of the features empirically and the results are presented in Appendix B.2. We present the Pearson correlation coefficient and Mutual Information between each feature and the class label, and the weight learned by linear SVM for each feature. Detailed discussions can be found in Appendix B.2.
>
> **2. A comparison with other state-of-art methods (not based on column generation) for VCP should be considered.**
>
> We didn’t compare our method with non-CG approaches in our experimental evaluation as the aim of this paper is to show that machine learning techniques can be developed to predict high-quality columns and to improve CG. However, as the vertex coloring problem has a decomposable structure that can be exploited by CG, we expect the CG-based approach is very effective in solving the vertex coloring problem compared with non-CG approaches. In addition, state-of-the-art branch-and-price based methods for this problem are highly complex collections of algorithms that work together to solve the problem. In this context it would be difficult to identify the contribution that any single element of the overall approach is making. By using this simpler approach, it is easier to both describe exactly how we are solving the problem and to demonstrate the effectiveness of the machine learning approach for improving CG.
>
> **3. Should the ≥ in (2) be equality?**
>
> For the VCP, since adjacent vertices cannot share the same color by the problem definition, the vertices that are of the same color in any feasible solution must form an Independent Set. Therefore, the VCP is equivalent to a Set Partitioning problem which aims to select the minimum number of Independent Sets from a graph such that each vertex is covered exactly once. This is also equivalent to a Set Covering problem, the objective of which is to minimize the number of Maximal Independent Sets (MISs) selected such that each vertex in the graph is covered at least once. In other words, if columns are defined as Independent Sets, Eq (2) should be equality constraint, and the formulation is known as set partitioning formulation. In our case which is the set covering formulation, the columns are defined as MISs, and Eq (2) is inequality constraint. A solution to the set covering formulation may assign multiple colors to a vertex, however using any of the assigned colors to a vertex is a feasible solution to the VCP. We have clarified this in Section 2 of our manuscript.
>
> **4. Can you add a reference for this statement that the set covering formulation (1)-(3) has a tight LP relaxation? Is it also true that, if you select a subset $\tilde{\mathbb{S}}$ of $\mathbb{S}$ and replace $\mathbb{S}$, the binary IP (1)-(3) still has a tight LP relaxation?**
>
> We have added the reference [1] to the statement. It can be shown that the LP relaxation of (1)-(3) provides a bound at least as good as that of the compact MIP formulation [1]. In addition, we observe that the lower bound produced by the LP relaxation of (1)-(3) is very tight for most of the problem instances tested in our experiments. It proves optimality for 18 out of 33 problem instances in Table B.5.
>
> The LP relaxation of (1)-(3) is a valid lower bound for the VCP only if the full set of MISs are considered. We believe that the binary IP (1)-(3) with a small subset of $\mathbb{S}$ selected does not necessarily have a tight LP relaxation.
>
> *[1] Anuj Mehrotra and Michael A Trick. A column generation approach for graph coloring. Informs Journal on Computing, 8(4):344–354, 1996.*
>
> **5. $\tilde{\mathbb{S}}$ should be $\tilde{\mathbb{S}}_v$ in constraint (5).**
>
> Thanks for spotting the typo. We have changed $\tilde{\mathbb{S}}$ to $\tilde{\mathbb{S}}_v$ and defined $\tilde{\mathbb{S}}_v$ in Section 2.
>
> **6. In Algorithm 1, you should either say "top $\epsilon$ of " or use some other notation such as $\kappa$.**
>
> We have replaced $\epsilon$ with $\kappa$ as suggested.

---

### Decision · Program_Chairs · 2023-01-20

**Decision:**

Accept: poster

**Justification For Why Not Higher Score:**

The paper makes an interesting contribution that bridges combinatorial optimization and machine learning. The idea itself is novel and interesting, but the techniques are somewhat limited.

**Justification For Why Not Lower Score:**

The overall approach is broadly applicable and it may lead to further progress.

**Metareview: Summary, Strengths And Weaknesses:**

The paper introduces an approach for integrating machine learning predictions with the column generation approach for solving combinatorial optimization problems. Specifically, the paper designs a set of features that can be used to predict whether a column belongs to an optimal integral solution, leading to machine learning augmented algorithms that construct integral solutions for combinatorial optimization problems.

The reviewers appreciated the approach of the paper and the novel integration of machine learning predictions with classical integer programming approaches for combinatorial optimization. Some of the reviewers were concerned about the general applicability of the proposed approach, and these concerns were satisfactorily addressed by the author response. The reviewers also felt that the main idea is interesting and novel but the techniques are somewhat limited.

**Note From Pc:**

if the above contains the word "oral" or "spotlight" please see: "oral" presentation means -> notable-top-5% and "spotlight" means -> notable-top-25%. As stated in our emails, we are disassociating presentation type from AC recommendations